# The instantaneous impact of calving and thinning on the Larsen C Ice Shelf

Tom Mitcham[1], G. Hilmar Gudmundsson[2], and Jonathan L. Bamber[1]

[1]Bristol Glaciology Centre, School of Geographical Sciences, University of Bristol, Bristol, UK
[2]Department of Geography and Environmental Sciences, Northumbria University, Newcastle, UK

**Correspondence:** Tom Mitcham (tom.mitcham@bristol.ac.uk)

**Abstract.** The Antarctic Peninsula has seen rapid and widespread changes in the extent of its ice shelves in recent decades, including the collapse of the Larsen A and B ice shelves in 1995 and 2002, respectively. In 2017 the Larsen C Ice Shelf (LCIS) lost around 10% of its area by calving one of the largest icebergs ever recorded (A68). This has raised questions about the structural integrity of the shelf and the impact of any changes in its extent on the flow of its tributary glaciers. In this work, we

used an ice flow model to study the instantaneous impact of changes in the thickness and extent of the LCIS on ice dynamics, and in particular on changes in the grounding line flux (GLF). We initialised the model to a pre-A68 calving state, and first replicated the calving of the A68 iceberg. We found that there was a limited instantaneous impact on upstream flow – with speeds increasing by less than 10% across almost all of the shelf – and a 0.28% increase in GLF. This result is supported by observations of ice velocity made before and after the calving event. We then perturbed the ice-shelf geometry through a series

of instantaneous, idealised calving and thinning experiments of increasing magnitude. We found that significant changes to the geometry of the ice shelf, through both calving and thinning, resulted in limited instantaneous changes in GLF. For example, to produce a doubling of GLF from calving, the new calving front needed to be moved to 5 km from the grounding line, removing almost the entire ice shelf. For thinning, over 200 m of the ice-shelf thickness had to be removed across the whole shelf to produce a doubling of GLF. Calculating the instantaneous increase in GLF (607%) after removing the entire ice shelf allowed

us to quantify the total amount of buttressing provided by the LCIS. From this, we identified that the region of the ice shelf in the first 5 km downstream of the grounding line provided over 80% of the buttressing capacity of the shelf. This is due to the large resistive stresses generated in the narrow, local embayments downstream of the largest tributary glaciers.

## 1   Introduction

Around 74% of the Antarctic coastline is fringed by floating ice shelves (Bindschadler et al., 2011). These ice shelves are fed

by tributary glaciers and ice streams, and lose mass predominantly through basal melting at the ice-ocean interface, and calving at the ice front (Depoorter et al., 2013). When formed in embayments, or where they locally run aground at ice rises or pinning points, ice shelves can generate resistive stresses which are transferred through the ice shelf to the grounding line (GL), where they provide a backstress to the grounded ice sheet (Thomas, 1979). This process, known as ice-shelf buttressing, means that

ice shelves can exert a mechanical control on the grounding line flux (GLF), and therefore control the rate at which the ice

sheet contributes to changes in global sea level (e.g. Dupont and Alley, 2005; Gudmundsson, 2013).

The Larsen C Ice Shelf (LCIS) is situated on the eastern side of the northern Antarctic Peninsula (AP), and is the fourth largest ice shelf in Antarctica. Over the second half of the 20th century, increasing surface air temperatures – and a subsequent increase in surface melt – have been implicated in the breakup of several ice shelves on the AP (e.g. Morris and Vaughan, 2003; Vaughan et al., 2003; Khazendar et al., 2011; Banwell et al., 2013). In 1995 the Larsen A ice shelf (LAIS) collapsed (Rott et al.,

1996), and in 2002, the Larsen B ice shelf (LBIS) disintegrated in a matter of six weeks (Scambos et al., 2004). Domack et al. (2005) showed from a subsequent analysis of marine sediments that the LBIS had been present for at least 12,000 years.

In 2017, the LCIS calved one of the largest icebergs ever recorded – named A68 – reducing its surface area by $\sim 10\%$. The ice-shelf extent is now at its minimum since satellite observations began (Hogg and Gudmundsson, 2017). During the 2019/20 austral summer, the LCIS experienced near-record levels of surface melting (Bevan et al., 2020). All of these factors

have raised questions about the future viability of the LCIS (e.g. Kulessa et al., 2014; Jansen et al., 2015; Holland et al., 2015), and what the consequences of any changes to its thickness or extent might be for the subsequent ice dynamics of the AP (e.g. Schannwell et al., 2018).

Following the collapse of the LBIS in 2002, a significant change in the flow of its tributary glaciers was observed, with some increasing in speed by close to 900% (Rignot et al., 2004). This increase in ice speeds – and consequently GLF – has

been sustained to the present day (Berthier et al., 2012; Rott et al., 2018). De Rydt et al. (2015) modelled the response to this rapid loss of ice-shelf buttressing through diagnostic (or time-independent) simulations with the ice flow model Úa, which is also used in this study. They were able to reproduce the spatial variability of the response in ice velocity across the tributary glaciers, but suggested that transient experiments would be required to simulate the observed, quantitative changes. A similar approach – also using Úa – was taken to model the response to the collapse of the LAIS (Royston and Gudmundsson, 2016).

This study found that the initial increase in GLF could be reproduced with diagnostic experiments, but that modelling the transient redistribution of mass was required to reproduce changes further upstream in the tributary glaciers.

A number of studies have previously examined buttressing on the LCIS. Borstad et al. (2013) modelled the stress field in the ice shelf, calculated a local buttressing number, and modelled the impact of removing basal contact at ice rises on the dynamics of the shelf, but not the tributary glaciers. Fürst et al. (2016) mapped the 'maximum buttressing' number across the ice shelf,

and from this delineated regions of 'passive ice' which could be calved without significantly increasing the ice flux across new calving fronts in the shelf. Reese et al. (2018) computed the impact of small perturbations in ice-shelf thickness on the integrated GLF, producing a map of the 'buttressing flux response number' across the shelf. This allowed them to determine the regions in the ice shelf where a perturbation in ice thickness would produce the largest response in GLF, and they also demonstrated that small changes in ice-shelf thickness could impact the GLF hundreds of kilometres away. Gudmundsson

et al. (2019) modelled the impact of an instantaneous thinning of Antarctic ice shelves on the grounded ice and GLF, with a spatial pattern and amplitude derived from observations. They highlight the fact that changes in ice-shelf buttressing have an instantaneous impact on ice velocities, after which there is a transient adjustment to the flow and a redistribution of mass. In their experiments, they were able to map, or 'fingerprint', the instantaneous ice velocity response and the reduction in

buttressing due to the cumulative, observed ice-shelf thinning from 1994 to 2017. Zhang et al. (2020) explored the correlation between locally derived buttressing numbers in the ice shelf, and changes in GLF due to small perturbations in ice-shelf thickness at the same locations. They found that for a real-world ice shelf (the LCIS) there was no relationship between these two measures, and that locally derived buttressing numbers are not predictors for the impact of perturbations in ice-shelf geometry on GLF. Finally, Schannwell et al. (2018) and Sun et al. (2020) explored the transient response of the grounded ice to the complete collapse of the LCIS, and the associated removal of all ice-shelf buttressing.

Here, we build on this existing literature through a series of diagnostic perturbation experiments, including ice-shelf calving and thinning and ungrounding from ice rises. This approach allows us to explore the buttressing capacity of the LCIS, due to the instantaneous impact that changes in buttressing have on the ice flow, but we do not examine the transient redistribution of mass in response to the perturbations. Our first objective is to model the response of the LCIS and its tributaries to the calving of the A68 iceberg and validate these results with observations. We then study the instantaneous GLF response to a series of idealised ice-shelf calving events. By quantifying the maximum GLF response, we determine the total amount of buttressing provided by the ice shelf, and examine the proportion of this total that is generated by different regions of the shelf. We simulate the loss of basal contact of the ice shelf at the Bawden and Gipps ice rises (outlined and labelled in Fig. 1), again examining the impact on GLF. And finally, we systematically perturb the thickness of the ice shelf by increasing amounts, again aiming to understand how much the ice-shelf geometry needs to change before a significant response in GLF is produced.

## 2 Methods

### 2.1 Ice flow model

We used the Úa ice flow model (Gudmundsson et al., 2012), which solves the vertically integrated, shallow-shelf approximation (e.g. MacAyeal, 1989) using the finite element method on an unstructured mesh. Úa has been used in both idealised (e.g. Gudmundsson et al., 2012; Gudmundsson, 2013) and realistic (e.g. De Rydt et al., 2015; Minchew et al., 2018; Hill et al., 2018; Reese et al., 2018; Gudmundsson et al., 2019) settings to examine the response of grounded ice to perturbations in the ice shelf. It has also been tested in recent model intercomparison projects (Pattyn et al., 2013; Cornford et al., 2020).

The equation solved in Úa for the vertically integrated balance of stresses is

$$\boldsymbol{\nabla}_{xy} \cdot (h\boldsymbol{R}) - \boldsymbol{t_{bh}} = \rho_i g h \boldsymbol{\nabla}_{xy} s + \frac{1}{2} g h^2 \boldsymbol{\nabla}_{xy} \rho_i, \tag{1}$$

where

$$\boldsymbol{R} = \begin{pmatrix} 2\tau_{xx} + \tau_{yy} & \tau_{xy} \\ \tau_{xy} & 2\tau_{yy} + \tau_{xx} \end{pmatrix} \tag{2}$$

is the resistive stress tensor, $\tau_{ij}$ are the components of the deviatoric stress tensor, $\boldsymbol{\nabla}_{xy} = (\partial_x, \partial_y)^T$, $\boldsymbol{t_{bh}}$ is the horizontal component of the basal traction, $h$ is the ice thickness, $s$ is the ice surface elevation, $\rho_i$ is the vertically integrated ice density (which varies spatially) and $g$ is the acceleration due to gravity.

In this work we conduct diagnostic – or time-independent – experiments, in which the equations for stress balance are solved together with Glen's flow law, the constitutive equation linking the stress field in the ice to deformation

$$\dot{\epsilon}_{ij} = A\tau^{(n-1)}\tau_{ij} \tag{3}$$

where $\dot{\epsilon}_{ij}$ are the components of the strain rate tensor, $\tau$ is the second invariant of the deviatoric stress tensor, given by

$$\tau = \sqrt{\tau_{ij}\tau_{ij}/2} \tag{4}$$

and the rate factor, $A$ – which depends on ice properties including temperature, crystal fabric and damage – was optimised using inverse methods (see Sect. 2.3). We set the creep exponent $n = 3$ as is standard in ice-flow modelling.

A non-linear Weertman type sliding law was used with the following form

$$\boldsymbol{t_{bh}} = C^{-1/m}||\boldsymbol{u_b}||^{(1-m)/m}\boldsymbol{u_b} \tag{5}$$

where $\boldsymbol{t_{bh}}$ is the horizontal component of the the bed-tangential basal traction and $\boldsymbol{u_b}$ the horizontal component of the bed-tangential ice velocity. The basal slipperiness parameter, $C$, is also inferred using inverse methods, and $m = 3$. The results of diagnostic perturbation experiments using this ice flow model have previously been found to be largely unaffected by the value chosen for $m$ (e.g. Hill et al., 2018; Gudmundsson et al., 2019). However, we conducted additional sensitivity tests to examine the impact of using different stress exponents ($m$ values) in the sliding law, and found that they do not affect the conclusions of our study. The details of these sensitivity tests are presented in Appendix E.

## 2.2 Model domain and data

The model domain, shown in Fig. 1, includes all of the drainage basins identified by Cook and Vaughan (2010) that drain into the LCIS. The calving front location represents a pre-July 2017 state, before the A68 iceberg calved from the shelf, and was defined as the maximum ice extent in the BedMachine Antarctica v2 data set (Morlighem et al., 2020). One artificial boundary was drawn to separate the region between the Larsen C and D ice shelves, and this was manually delineated by joining the ice divide to the calving front.

The finite element mesh used in the computation was generated with the open source Gmsh software (Geuzaine and Remacle, 2009). We chose to use linear shape functions on these elements. The target element size was set to 2 km across the floating ice shelf, with the resolution increased around the grounding line, where elements 250 m in size were used. The mesh was refined to 1 km in all tributary glaciers, and 500 m in regions of high strain rates. This ultimately produced a mesh with $\sim 154,000$ elements with a maximum, median, and minimum element size of 4,000 m, 640 m, and 160 m respectively. The dependence of the model results on element size was tested in a convergence analysis and the effect was found to be negligible. The results of these convergence tests are shown in Appendix C.

Along the ice divides at the boundary of the model domain, a zero velocity boundary condition was applied. At the calving front, a stress boundary condition, arising from ocean pressure, was prescribed.

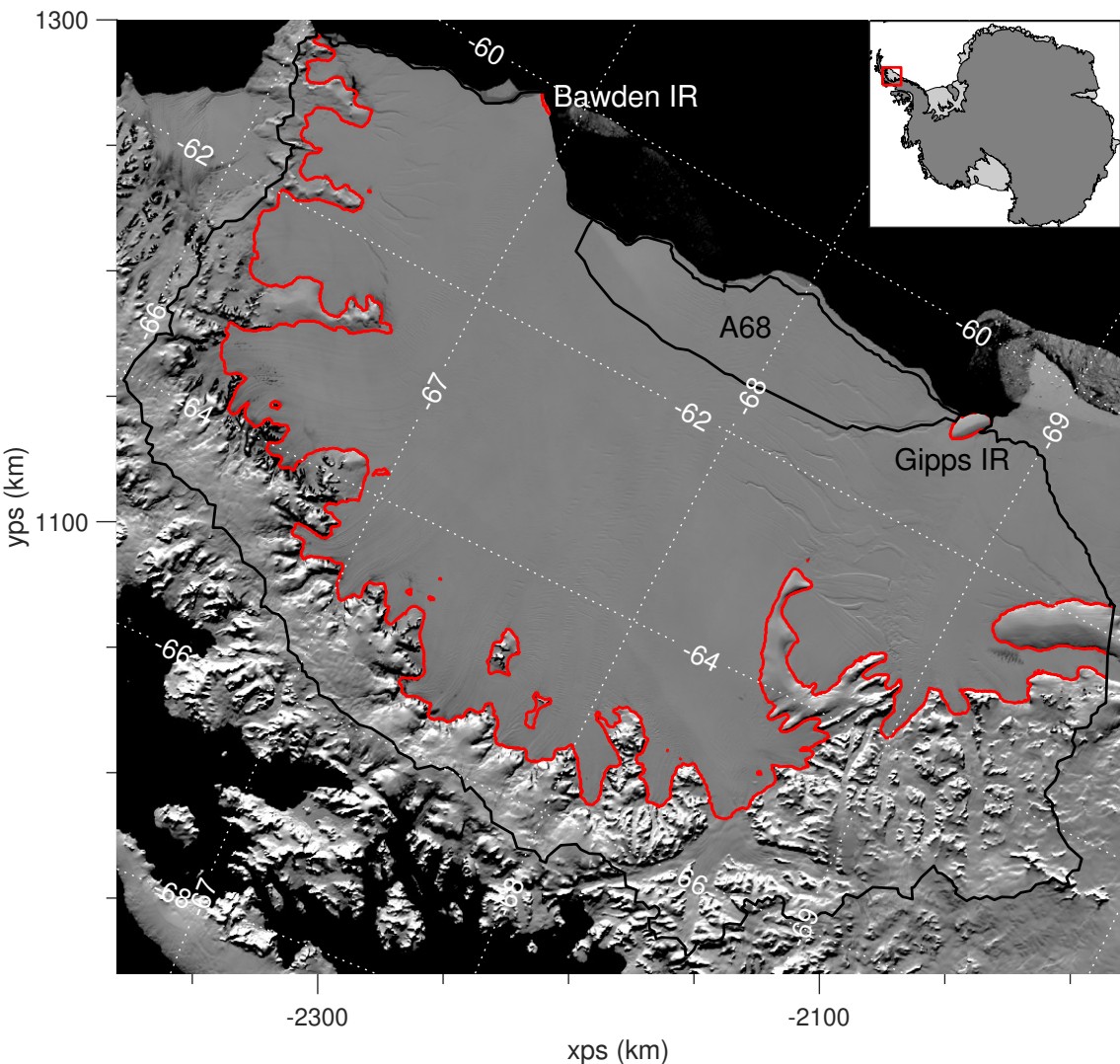

**Figure 1.** MODIS Mosaic of Antarctica (Scambos et al., 2007) image of the Larsen C Ice Shelf and its tributary glaciers, with open ocean shown in black. The boundary of the model domain is plotted in black, with the outline of the A68 iceberg which calved in July 2017 also shown. In red is the grounding line position as calculated in the model, which remains fixed throughout the experiments. The outlines of the Bawden and Gipps ice rises at the calving front of the ice shelf are also plotted in red. Lines of latitude and longitude are shown in the white, dotted lines. The coordinate system used here – and in all other maps – is the WGS84 Antarctic Polar Stereographic projection (EPSG:3031), and the axes labels (also used in all other maps) refer to the x and y directions in these polar stereographic coordinates.

The initial ice thickness, surface elevation and bedrock topography was taken from the BedMachine Antarctica v2 data
set (Morlighem et al., 2020). The surface elevation was adjusted at a few points (in areas of exposed bedrock) to ensure that
at least 1 m of ice was present across the whole computational domain. The firn air content field provided in this data set

is derived from a firn densification model (Ligtenberg et al., 2011), which was forced by RACMO2.3p2 at the surface (van Wessem et al., 2018). This field was used to calculate a spatially variable, depth-integrated ice density across the model domain. Due to discrepancies between the ice thickness and firn air content fields, it was necessary to introduce a minimum value of 800 $\mathrm{kgm}^{-3}$ for the depth-integrated ice density. A map of the resulting ice-density field is shown in Fig.D1. The impact of using a horizontally spatially variable ice density – as opposed to a constant ice density of 917 $\mathrm{kgm}^{-3}$ – on our results is minimal. The details and results of the sensitivity tests undertaken to determine this are outlined in Appendix D.

For the optimisation procedure used to initialise the model (see Sect. 2.3) we used the MEaSUREs InSAR-based Antarctic Ice Velocity v2 data set (Rignot et al., 2017, 2011; Mouginot et al., 2012). For model validation (see Sect. 3.1) we used ice velocity measurements generated from Sentinel-1 SAR data which were provided by ENVEO. This data set consisted of monthly maps of tide-corrected ice velocities over the LCIS and its tributaries from October 2014 - September 2019.

## 2.3  Model initialisation

To generate the initial conditions the rate factor, $A$, and basal slipperiness parameter, $C$, were optimised by minimising the misfit between observed and modelled ice velocity through inverse methods widely used in glaciology (e.g. MacAyeal, 1993). The cost function that is minimised in Úa during the optimisation is $J = I + R$, where

$$I = \frac{1}{2\mathcal{A}} \int \left( \frac{\boldsymbol{u} - \boldsymbol{u_{obs}}}{\boldsymbol{u_{err}}} \right)^2 d\mathcal{A}, \tag{6}$$

is the misfit term, and

$$R = \frac{1}{2\mathcal{A}} \int \left[ \gamma_{sA}^2 \left( \nabla \log_{10} \left( \frac{A}{\hat{A}} \right) \right)^2 + \gamma_{sC}^2 \left( \nabla \log_{10} \left( \frac{C}{\hat{C}} \right) \right)^2 + \gamma_{aA}^2 \left( \log_{10} \left( \frac{A}{\hat{A}} \right) \right)^2 + \gamma_{aC}^2 \left( \log_{10} \left( \frac{C}{\hat{C}} \right) \right)^2 \right] d\mathcal{A}, \tag{7}$$

is the regularisation term. Here, $\mathcal{A}$ is the area of the model domain and $\boldsymbol{u}$ represents the $x$ and $y$ components of the modelled ice velocity. $\boldsymbol{u_{obs}}$ and $\boldsymbol{u_{err}}$ represent the components of the measured surface ice velocity and their associated uncertainties, both here taken from the MEaSUREs InSAR-based Antarctic Ice Velocity v2 data set. $\gamma_{sA}$, $\gamma_{sC}$, $\gamma_{aA}$ and $\gamma_{aC}$ are regularisation parameters that penalise deviations in the fields being optimised – in this case $A$ and $C$ – from their prior estimates – $\hat{A}$ and $\hat{C}$ – in terms of gradient and amplitude respectively. The priors ($\hat{A}$ and $\hat{C}$) were chosen to be spatially uniform. The value chosen for $\hat{A} = 1.15 \times 10^{-8}$ $\mathrm{a}^{-1}\mathrm{kPa}^{-3}$, which corresponds to ice at a temperature of $-10°C$ as given by the equation for the rate factor in Morland and Smith (1984). The value for chosen for $\hat{C} = 1.95 \times 10^{-4}$ $\mathrm{ma}^{-1}\mathrm{kPa}^{-3}$, which was calculated from the sliding law (Eq. 5) assuming a basal shear stress of 80 kPa and an ice velocity of 100 $\mathrm{ma}^{-1}$. We used $\gamma_{sA/C} = 1000$ and $\gamma_{aA/C} = 1$, with the four regularisation parameters determined using L-curve analyses (see Appendix A).

The resulting ice velocity field is shown in Fig. 2b. There was a good fit to the observed velocities across the domain, with a spatially averaged RMS difference between the observed and modelled velocities of 11.2 $\mathrm{ma}^{-1}$, and a particularly good fit across the ice shelf and at the GL. One region in which the model struggled to replicate the ice velocities is just to the north of the Gipps Ice Rise, where the nascent A68 iceberg was beginning to detach from the shelf. This gave rise to large strain rates in the shelf, which weren't captured in the model, presumably due to the regularisation applied to the rate factor, $A$. The resulting maps of $A$ and $C$ are shown in Fig. A2. Following the inversion, the modelled GLF across the main GL (i.e. excluding

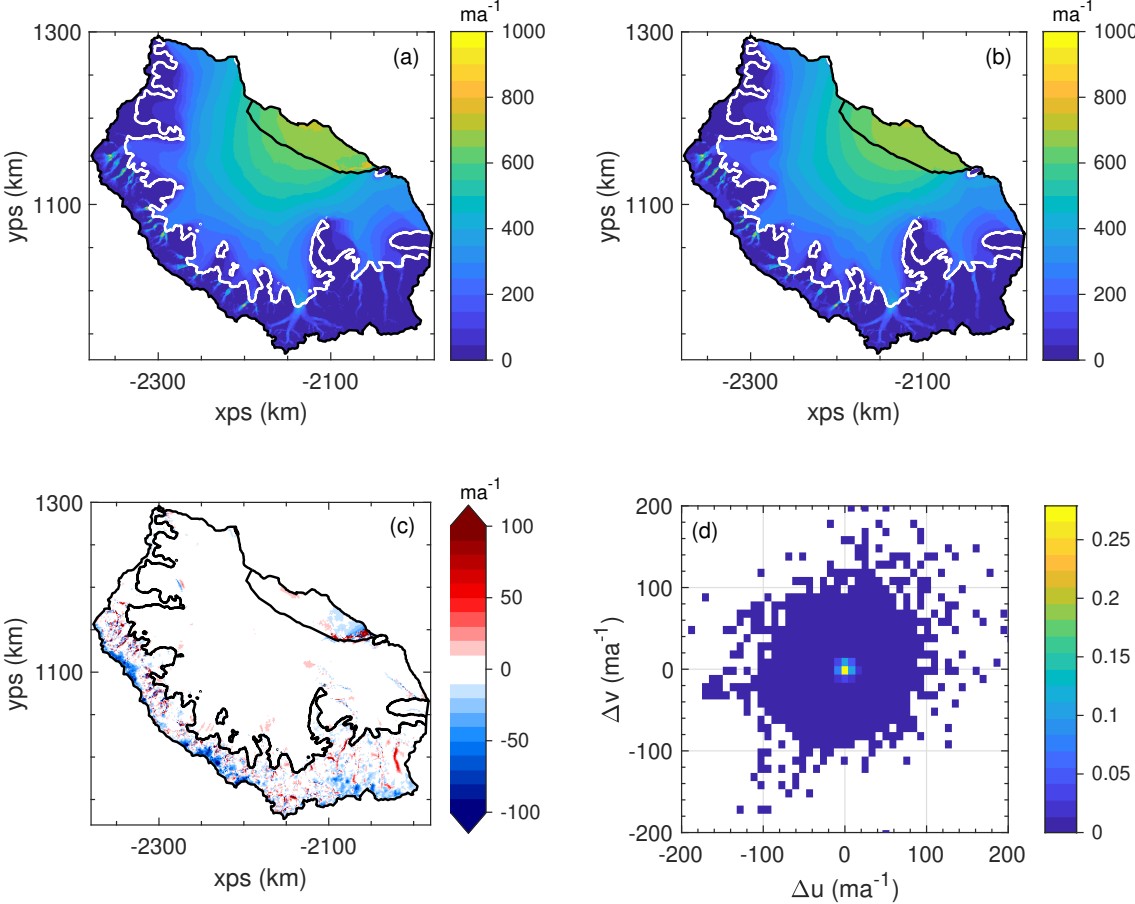

**Figure 2.** Observed ice speed **(a)** (from the MEaSUREs InSAR-based Antarctic ice velocity map) and modelled ice speed **(b)** following the initialisation procedure. Panel **(c)** shows the difference between the observed and modelled ice speed (obs - mod) and **(d)** is a normalised, bivariate histogram of the differences between the x and y components of the observed and modelled ice velocities at each node in the computational mesh.

fluxes across ice rises in the shelf) was $23.2 \ \mathrm{Gta}^{-1}$, and this was the reference GLF to which GLF changes in perturbation experiments were compared.

## 2.4 Calving experiments

From the initial conditions derived from the initialisation procedure, the geometry of the ice shelf was perturbed whilst holding all other parameters constant, generating a new stress field. This yielded an instantaneous change in the modelled ice velocity, which we compared to the initial velocity field and from which we calculated changes in GLF.

The first experiment undertaken was to replicate the calving of the A68 iceberg, the extent of which was derived from Landsat 8 images. To perform the experiment, the mesh elements within the region that calved were removed, thereby relocating

the model boundary to the new calving front. Through this procedure, the remaining elements of the mesh were left unchanged and any interpolation errors avoided.

In addition to the A68 calving event, a series of idealised calving experiments were conducted. The calving front was moved progressively nearer to the GL by removing mesh elements using a 'distance from the main grounding line' metric (mapped in Fig. 4c).

It is important to note that in these calving experiments (and indeed in all experiments in this study) no perturbation was applied to the nodal values of any element which crossed the main GL. This meant that there was no change in driving stresses across the GL, and that the GL location remained fixed in all experiments. This ensured that any change in GLF was due solely to changes in the buttressing provided by the ice shelf.

## 2.5 Ungrounding and thinning experiments

To simulate the ungrounding of the LCIS from the Bawden and Gipps ice rises, the bed topography was lowered so that the ice shelf became afloat without changing the ice thickness. This experiment was carried out for the ungrounding from the two ice rises individually, and then for a 'combined' ungrounding, in which both contacts were removed simultaneously.

Finally, we explored the GLF response to perturbations in ice thickness. The ice thickness at nodes belonging to elements that were fully afloat (again, to ensure no change in driving stress across the GL) was progressively reduced until the whole ice shelf had a thickness of only $1\,\mathrm{m}$. The $1\,\mathrm{m}$ thin layer of ice across the shelf was maintained for computational reasons, and the results are insensitive to a further reduction in the minimum ice thickness. Locally this thinning was done both proportionally, i.e. the ice thickness was reduced by a given fraction of the total thickness at each node, and uniformly where thickness was reduced across the whole ice shelf by the same fixed amount. The subsequent changes in ice velocity and GLF were calculated for each step in the series of experiments.

## 3 Results

We first present the results of the A68 iceberg calving experiment, before showing the changes in GLF in response to the idealised calving experiments. We then examine the ice flow response to the ungrounding of the Bawden and Gipps ice rises, before presenting the results of the ice-shelf thinning perturbations as outlined in the previous section. As previously stated, all of the modelling experiments conducted in this study are diagnostic (or time-independent) and therefore the changes in ice flow and GLF presented here are instantaneous changes.

### 3.1 A68 calving

In response to the removal of the A68 iceberg from the model domain, there was an instantaneous increase in ice velocity immediately upstream of the new calving front of up to $\sim 100\,\mathrm{ma^{-1}}$ (Fig. 3a). The spatial extent of this velocity response was limited, and across almost all of the ice shelf the change in velocity was smaller than 10% – even becoming negative in the

region of the shelf to the north of the Gipps Ice Rise. The changes in velocity did not extend throughout the whole ice shelf, and as such there was almost no modelled increase in GLF (0.28%) due to this calving event.

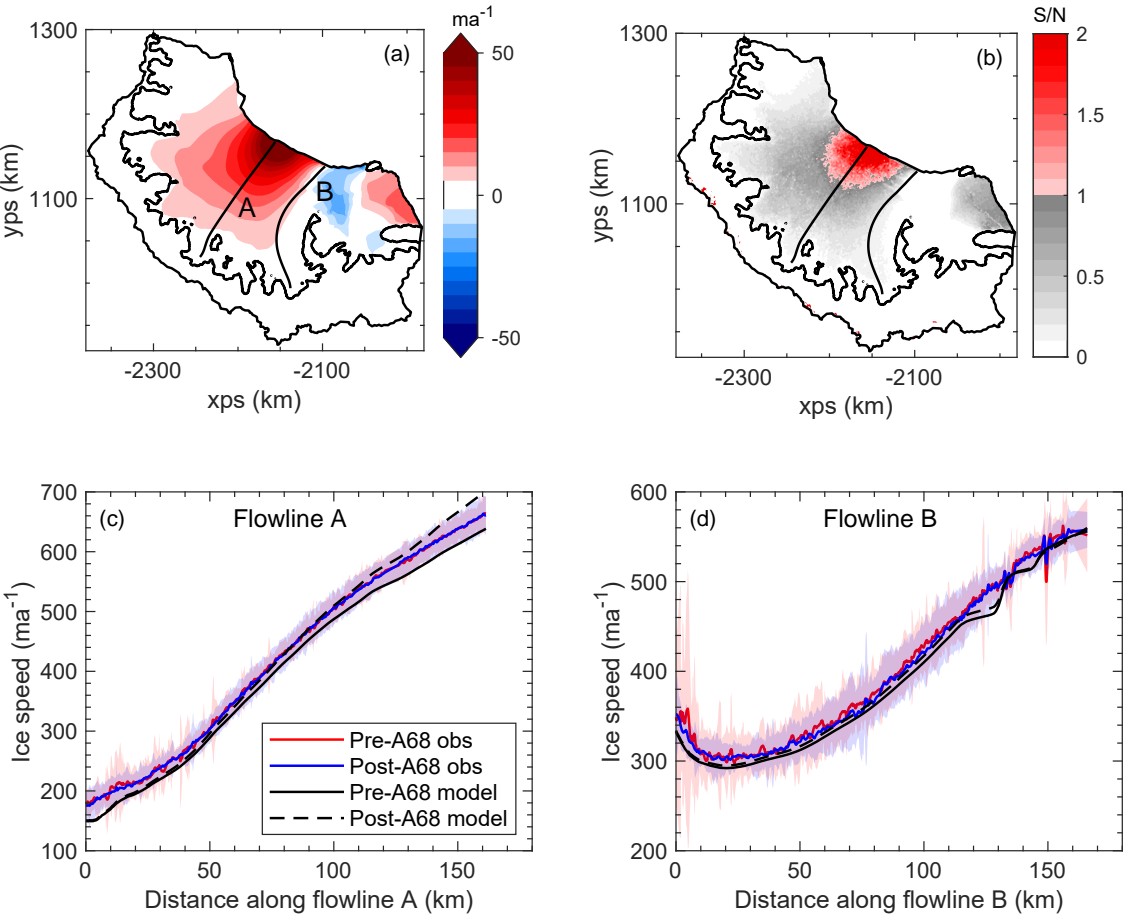

**Figure 3.** The modelled change in ice speed due to the calving of the A68 iceberg **(a)** along with the paths of the two flowlines used. **(b)** is the ratio of the 'signal' (model response in panel (a)) to the 'noise' ($2\sigma$ variability in the monthly observations of ice velocity). Panels **(c)** and **(d)** show the modelled ice speed before and after the calving event along two flowlines in the shelf. The mean, observed speed before and after July 2017, together with shading representing the $2\sigma$ variability, is also plotted.

We compared our model results to ice velocity observations produced and provided by ENVEO from analysis of Sentinel-1 
SAR data. From this monthly time series of ice velocity maps, we calculated the mean and standard deviation in ice velocity at each point in the model domain for both the October 2014 - June 2017 (pre-A68) and August 2017 - September 2019 (post-A68) periods (Fig. 3).

By plotting the 'signal' (the modelled response to A68 calving) to 'noise' ($2\sigma$ variation in observations) ratio in Fig. 3b, and examining the data along two flowlines on the shelf (Fig. 3 c and d), we see that the modelled response largely falls within the 
200 internal variability of the ice velocity in the shelf. We also see that the mean ice velocity before and after the A68 calving event

are nearly identical, demonstrating that during this five year measurement window there has been no observable change in ice velocity, and no transient or sustained response to the calving of the A68 iceberg.

## 3.2 Idealised calving experiments

The impact of moving the calving front progressively closer to the GL on the GLF can be seen in Fig. 4a and b. It shows that a
retreat of the calving front from its present day position back into the embayment produced a limited instantaneous impact on the GLF. The calving front had to be retreated to 40 km from the GL to induce a 10% increase in GLF. For a doubling of GLF, the calving front had to be positioned 5 km from the grounding line, removing almost all of the ice shelf in the process.

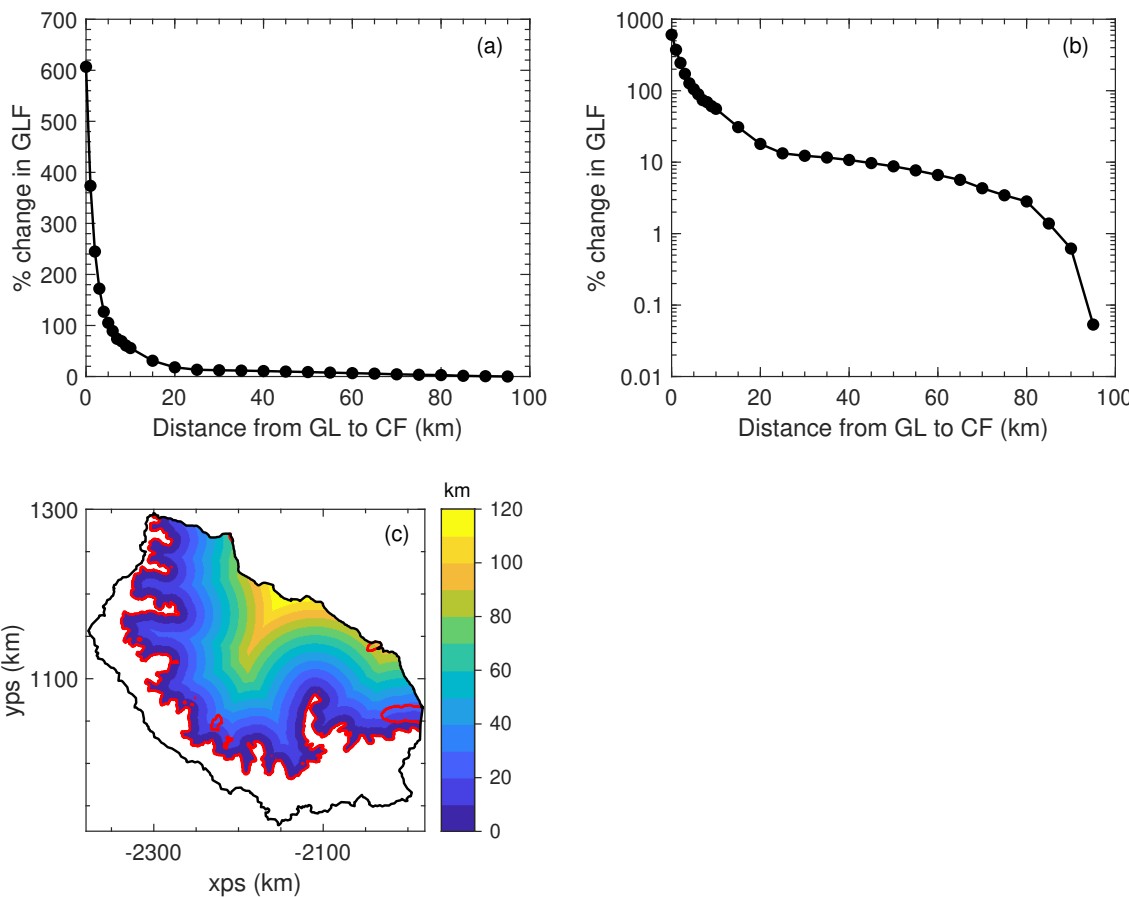

**Figure 4.** Calving experiments: **(a)** is the percentage change in grounding line flux for the each idealised calving experiment, and **(b)** shows the same data, but with a log scale on the y-axis. A map of the 'distance to the main grounding line' metric, used to define the calving experiments, is shown in **(c)**, where the red line is the grounding line and the black line the boundary of the model domain.

The maximum GLF increase (607%) – from the complete removal of the ice shelf – can be thought of as representing the total buttressing provided by the LCIS in its current configuration to its grounded tributary glaciers. By comparing the instantaneous

increase in GLF for each idealised calving experiment to the maximum GLF increase from complete ice-shelf removal, we are able to calculate the proportion of the total buttressing that remains after each perturbation experiment. Therefore, Fig. 4a and b show what proportion of the total buttressing is provided by each section of the ice shelf removed in the series of calving perturbations. From this, we see that over 95% of the total buttressing is provided by ice in the first 25 km downstream of the GL, and that over 80% is generated in the first 5 km of ice immediately downstream of the GL.

### 3.3 Ungrounding experiments

In the modelled response to the ungrounding of the LCIS from the Bawden ice rise there was a significant local instantaneous increase in velocity upstream of the ice rise (Fig. 5a) of $\sim 200$ ma$^{-1}$ (and an even greater increase for the ice that was previously grounded). This represents a $\sim 50\%$ increase in ice velocity in this region. However, as with the A68 calving, this instantaneous velocity response is spatially limited and there is a just a 1% increase in GLF from this perturbation.

A similar, localised response in ice velocity is seen when the Gipps Ice Rise contact is removed (Fig. 5b), and a similar instantaneous increase in GLF (1.2%) is modelled. Fig. 5c shows the ice velocity response to the simultaneous loss of contact from both ice rises, which produced an increase in velocity across the whole ice shelf. In Fig. 5d we show the difference between the combined ungrounding event and the sum of the two individual events. It shows that the combined ungrounding is approximately a linear superposition of the two individual events, and the corresponding change in GLF from the combined event is 2.2%. These experiments show us that the two ice rises provide a very small proportion of the total buttressing of the LCIS, but cannot tell us about the transient mass redistribution in response to the loss of basal contact at these locations.

### 3.4 Thinning experiments

The changes in GLF due to perturbations in the ice-shelf thickness are shown in Fig. 6 (the two different approaches to applying ice-shelf thinning are set out in Section 2.5). When thinning the ice shelf in the 'uniform' sense (Fig. 6a) we find that $\sim 30$ m of ice-shelf thinning is required to produce a 10% increase in GLF and that over 200 m of thinning is required to produce a doubling of GLF. 200 m of 'uniform' ice-shelf thinning removes 9050 Gt of ice from the LCIS. The equivalent idealised calving experiment – in terms of ice-mass removed – positions the calving front 15 km downstream from the GL, which only increased the GLF by around 30% (that calving experiment removed 9350 Gt of ice from the shelf).

The initial ice thickness across the model domain is shown in Fig. 6d. The maximum ice thickness at a computational node in the shelf was 1,489 m. Therefore, by applying a thinning perturbation larger than this (of 1,500 m), the ice shelf was reduced to the minimum thickness of 1 m everywhere after an algorithm in the model ensured that the minimum ice thickness was present everywhere in the model domain. However, in its initial state the median ice-shelf thickness was 290 m, 89% of the ice shelf had a thickness of less than 500 m and 99% less than 800 m. Therefore, the gradient of the curve in Fig. 6a decreases after $\sim 500$ m of applied thinning as areas of the shelf already at the minimum thickness are not affected by further increases in perturbation amplitude.

Fig. 6b shows the response in GLF to the 'proportional' thinning experiments. Here, 7% of the ice-shelf thickness needed to be removed to produce a 10% increase in GLF, and 45% removed to produce a doubling of GLF.

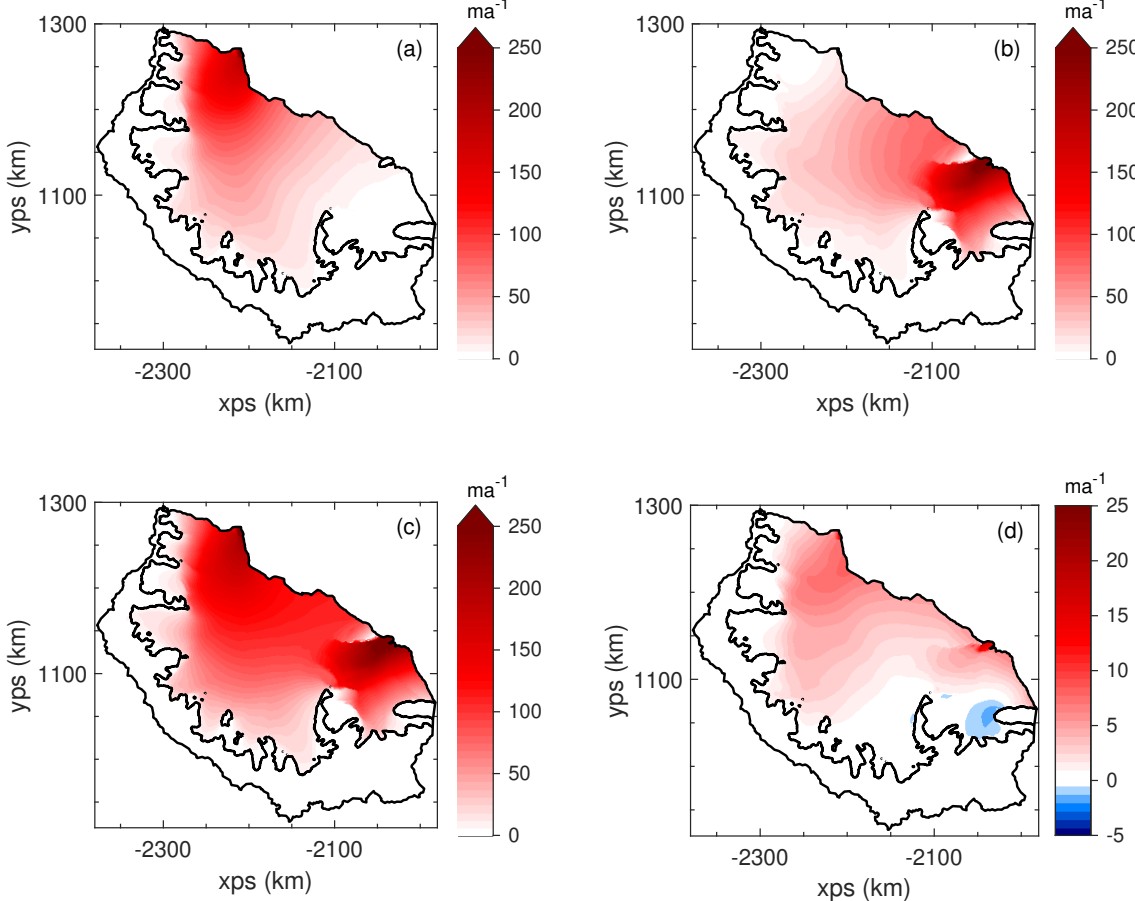

**Figure 5.** Ungrounding experiments: The change in modelled ice speed after ungrounding at the Bawden Ice Rise **(a)**, the Gipps Ice Rise **(b)** and at both ice rises simultaneously **(c)**. The difference between the combined ungrounding and the sum of the two individual ungrounding experiments is shown in **(d)**. Note the different scale on the colour bar in **(d)**. In panels **(a)**-**(c)** there is a small speed decrease (typically $< 10\ \mathrm{ma}^{-1}$) at a few computational nodes that is not visible when plotted, and therefore the colour scale has been truncated at zero for clarity.

By calculating the ice-shelf mass removed in each experiment we were able to compare the two approaches (Fig. 6c). We see from the 'uniform' experiment curve, that there is a large response in GLF to a small change in mass removed towards the

end of the series of perturbations, when the thickest ice is being significantly perturbed. The maximum GLF increase of 502% is identical in both the 'uniform' 1,500 m thinning and the 'proportional' 100% thinning experiments, as expected. The initial linear regimes in both sets of thinning perturbations are discussed in Appendix B.

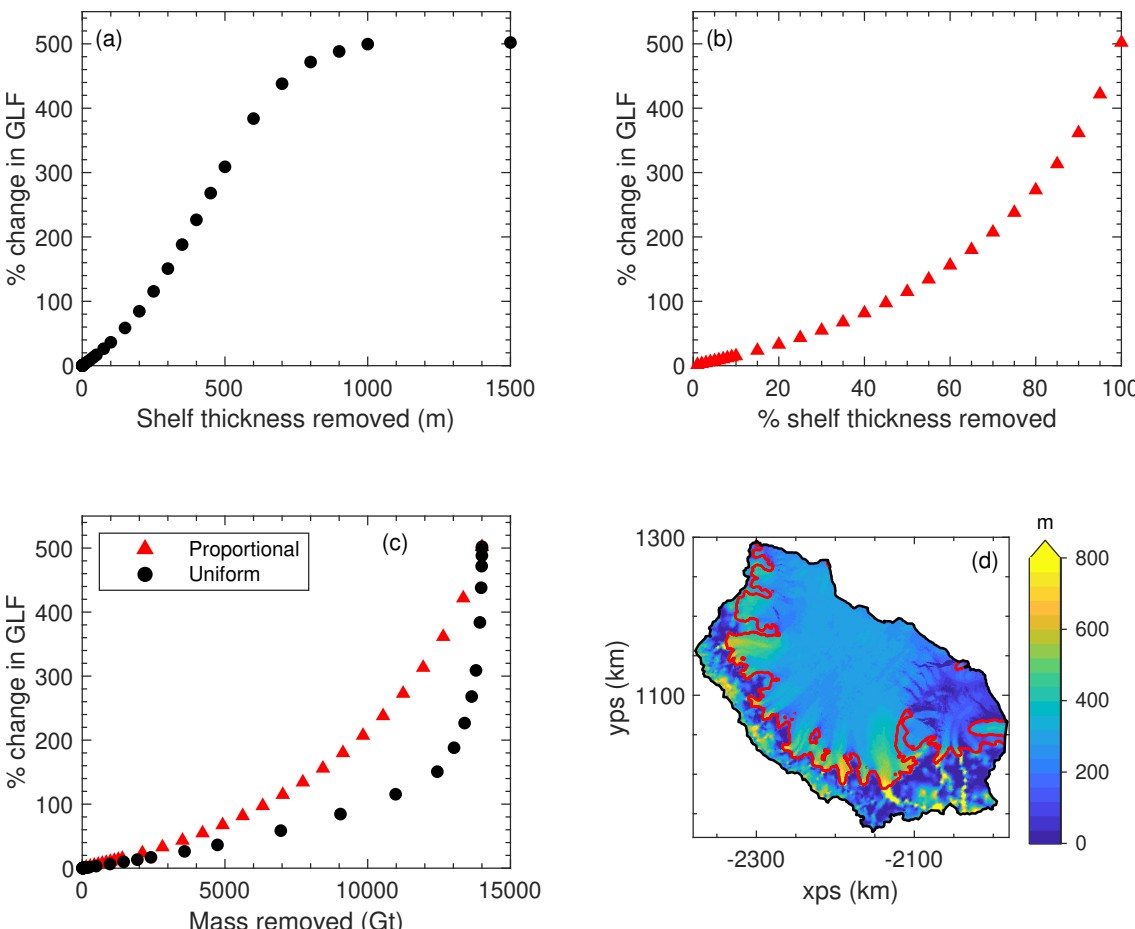

**Figure 6. (a)** is the percentage change in grounding line flux during the 'uniform' thinning experiments, and **(b)** is the same but for the 'proportional' thinning experiments. **(c)** is the response in grounding line flux for the two sets of experiments, but with the perturbations expressed in the amount of mass removed rather than the amount of thinning applied. **(d)** shows a map of ice thickness across the model domain from BedMachine Antarctica v2 (Morlighem et al., 2020), with grounding lines plotted in red.

## 4   Discussion

### 4.1   Calving experiments and ice-shelf buttressing

Our first objective was to model the instantaneous response of the LCIS to the calving of the A68 iceberg and compare the results to observations. The limited change in ice-shelf velocities, and lack of change in the GLF, suggests that this part of the ice shelf provided almost no buttressing. This finding is in agreement with the work of Fürst et al. (2016), who classified this region as 'passive ice', and with the map of 'buttressing flux response number' that Reese et al. (2018) produced. Borstad et al. (2017) hypothesised four potential calving events based on the trajectory in which the rift that eventually formed the A68

iceberg was growing, and modelled the response to these events. Their 'Scenario 2' is most similar to the calving event that eventually occurred, and our modelled results are in close agreement with theirs in both spatial pattern and amplitude (see Fig. 2d in Borstad et al., 2017).

The model results shows a decrease in ice velocity following the calving event in the region of the shelf just to the north of the Gipps Ice Rise. This is likely to be an artefact of the method used to perform the experiment. The rift that eventually formed the A68 iceberg had been present in the ice shelf for over a decade, and grew significantly during 2014 and 2016 (Jansen et al., 2015; Borstad et al., 2017). Therefore, the dynamic response to the detaching of the nascent A68 iceberg will have already taken place in this region, and this response is included in the ice velocity data used to initialise our model. Finally, in the model we essentially force the already detaching iceberg to have contact with ice upstream, inducing an artificial 'pulling' effect on this upstream ice, which is removed when the iceberg is calved from the domain. Evidence of this can be seen in the larger misfit between observed and modelled ice velocities in this region in Fig. 2c.

When comparing our result with observations, we found that the modelled response was smaller than the internal variability in the monthly ice velocity data available. Consequently, we were unable to validate the details in spatial pattern and amplitude of our modelled response. However, the observations do show that the mean ice velocity across the shelf before and after the calving event remained unchanged, or at least smaller than the measurement errors. This demonstrates that the calving of the A68 had little or no dynamic impact on the system, supporting ours and previous work that predicted such a response, or rather lack thereof.

The second aim of this work was to understand the instantaneous response in GLF to the migration of the calving front back towards the GL, and from that learn about the spatial distribution of the buttressing capacity of the ice shelf. Here we found that as the calving front is moved from its pre-A68 location to 25 km from the GL, there is an instantaneous increase of just 13% in GLF. It is only when regions of the shelf within 25 km of the GL are calved that ice in the narrow embayments downstream of the main tributary glaciers is removed, and this is where the increasing response in GLF begins. Calving perturbations up to this point remove more than 50% of the total ice-shelf mass, yet only induce an increase of 13% in the GLF.

Fürst et al. (2016) measured the impact of calving on the ice flux across the new marine ice front, not the GL, and therefore arrived at a different picture of buttressing on the LCIS. In their supplementary material, the impact of the same experiments on ice discharge across the GL was examined, and by that definition it was found that much more of the ice-shelf area was 'passive'. We argue that it is this second definition, the integrated impact of changes in ice-shelf geometry on stresses at the GL and consequently on instantaneous changes in GLF, that is the key measure of the buttressing capacity of ice shelves in their present configurations. This definition is in line with the original view of buttressing as the backstresses produced by an ice shelf that are felt at the grounding line (Thomas, 1979). This is the approach used in the work of Reese et al. (2018), Gudmundsson et al. (2019) and Zhang et al. (2020), who focus on the GLF response to ice-shelf perturbations. We find that the regions with the largest 'buttressing flux response number' (Reese et al., 2018) correspond to the regions providing the majority of the buttressing in our calving experiments. How the system redistributes mass transiently in response to a change in ice-shelf buttressing is an important question. But the diagnostic (time-independent) approach used here is sufficient to

reveal the buttressing capacity of the LCIS. Changes in buttressing produce an instantaneous response in ice velocities, and consequently in the GLF, as they are determined by the current state of stress in the ice.

The previous work of Reese et al. (2018) and Zhang et al. (2020) measured the response in GLF to small perturbations in ice-shelf geometry, but by removing the entire ice shelf and calculating the instantaneous response in GLF we are able to quantify the total amount of buttressing that the LCIS provides. This allowed us to examine what proportion of the total buttressing capacity is provided by different regions in the ice shelf. We find that over 95% of the buttressing is generated by the ice in the first $25\,\mathrm{km}$ downstream of the GL, and that over 80% comes from the first $5\,\mathrm{km}$ of ice directly downstream of the GL. The primary reason for this is that the LCIS geometry is characterised by a number of small, narrow embayments where the main tributary glaciers flow into the ice shelf. It is in these regions that the largest resistive stresses are generated, which dominate the buttressing capacity of the shelf as a whole.

Borstad et al. (2013), Fürst et al. (2016), Reese et al. (2018) and Zhang et al. (2020) find either elevated buttressing numbers or increased sensitivity of the GLF to ice-shelf thickness perturbations in the regions of the ice shelf upstream of the Bawden and Gipps ice rises. However, we find that calving these regions from the ice shelf does not produce a significant change in the GLF. This highlights the different conceptual and methodological approach to assessing ice-shelf buttressing that we use in this study. Whilst the GLF has previously been found to be sensitive to changes in ice-shelf geometry in these regions of the LCIS, we find that they do not contribute a significant amount to the buttressing of the grounded ice when the total buttressing capacity of the shelf as a whole is considered.

## 4.2 Ice-shelf thinning and ungrounding

We also set out to examine the GLF response to ice-shelf thickness perturbations. Fig. 6a shows that the response in GLF to thinning is approximately linear as a function of the amplitude of the thickness perturbation, as long as the amplitude is less than about $100\,\mathrm{m}$ (this is further explored in Appendix B). For amplitudes larger than about $100\,\mathrm{m}$, the response becomes progressively more non-linear, something that is also very evident when the GLF response is plotted as a function of the ice-shelf mass removed (Fig. 6c). Our explanation for this pattern of GLF response is that the thickest ice in the shelf, which is only reduced to the minimum ice thickness of $1\,\mathrm{m}$ towards the end of the series of 'uniform' experiments, is located directly downstream of the grounding line, where the largest tributary glaciers feed into the shelf. From our calving experiments, we saw that this is where the majority of the total buttressing capacity of the ice shelf is concentrated, and therefore the largest changes in GLF are seen when these regions of the shelf are thinned significantly.

The maximum increase in GLF due to the thinning experiments does not equal that of the calving experiments (502% vs 607%), as in the thinning experiments a $1\,\mathrm{m}$ thick layer of ice remains across the ice shelf. This means that at the new calving front the ice thickness is linearly interpolated between the unperturbed nodes in the computational mesh and the neighbouring nodes with an ice thickness of $1\,\mathrm{m}$. In the calving experiments mesh elements downstream of the new calving front are removed from the computational domain. We attribute the discrepancy between the maximum GLF increases to this difference in the numerical implementation of the calving and thinning experiments, and not to any residual buttressing effect of the $1\,\mathrm{m}$ ice layer. The minimum ice thickness is maintained across the computational domain for numerical reasons only, and we performed

sensitivity tests to determine the influence of different minimum ice thickness values. We find that increasing the minimum ice thickness to 10 m reduces the maximum GLF response from ice-shelf thinning to 475%, whilst reducing it further to 0.001 m,

only increases the maximum GLF response to 505% from the 502% modelled with a 1 m minimum ice thickness.

The instantaneous response to removing the basal contacts at the Bawden and Gipps ice rises has previously been modelled with different methods to ours. Borstad et al. (2013) only modelled the floating ice shelf, and therefore simulated a loss of contact at the ice rises by manually adjusting their inferred ice-viscosity parameter (the equivalent of our rate factor, $A$). The spatial pattern and amplitude of the increases in ice-shelf velocities closely matches those found in our experiments, and they

also found that the increase in ice-shelf velocities did not extend all the way upstream to the grounding line. Fürst et al. (2016) modelled the grounded ice as well as the shelf, and chose to set their basal friction coefficient to zero at the ice rises – removing the basal traction – rather than adjusting the ice or bed geometry. Despite these differences in approach, we found that our results are very similar in both spatial pattern and amplitude to these previous studies. In the experiment in which both ice rises were removed the instantaneous change in GLF was 2.2%. This suggests that, whilst these two ice rises may exert a

significant control on the flow of the shelf upstream of the pinning points, they do not exert a strong mechanical control on the ice flux at the GL, and only contribute a small amount to the total buttressing capacity of the shelf, given that their removal only affected the stresses at the GL enough to raise the GLF by 2.2% instantaneously.

The experiments conducted here are highly idealised in nature. We only considered 'uniform' or 'proportional' thinning perturbations to the shelf, and calving front locations were determined by a 'distance to the grounding line' metric. This

resulted in some unlikely calving front positions. More realistic calving experiments, using a physically based calving law or metric, could be used to model the response to more plausible ice-shelf configurations.

Secondly, this work only considered the instantaneous response to perturbations in ice-shelf thickness and extent. As changes to buttressing – through changes in the stresses in the ice – is an inherently instantaneous process, this approach is appropriate and sufficient to explore the buttressing capacity of the LCIS in its present state. However, the transient mass redistribution

in response to these perturbations, in which the acceleration would induce thinning and grounding line migration, requires further study and a different modelling approach. Of particular interest is whether the instantaneous GLF increases are the peak response to the perturbation, which then attenuates, or whether the GLF response is increased further by the transient evolution of the ice geometry.

## 5 Conclusions

In this study we have examined the instantaneous response of the LCIS and its tributaries to both observed and idealised perturbations to the ice-shelf geometry. We found that the calving of the A68 iceberg in July 2017 produced a limited change (mostly < 10%) in ice velocities in the shelf and had almost no instantaneous impact (a 0.28% increase) on the GLF. This finding is supported by observations which show no evidence of a change in velocity due to the calving event, and this furthermore confirms earlier work that suggested that the region that calved was largely 'passive ice'.

Through further, idealised calving experiments we found that a significant retreat of the calving front to 25 km downstream of the GL (removing over 50% of the ice-shelf mass) only produced a 13% instantaneous increase in GLF. Further retreat of the calving front to 5 km from the GL was needed to produce a doubling of GLF. By calculating the total buttressing provided by the LCIS – through modelling the instantaneous increase in GLF due to a complete collapse (607%) – we deduced that over 95% of the buttressing capacity of the LCIS is provided by ice within 25 km of the GL, in the narrow embayments downstream of the main tributary glaciers. We further found that over 80% of the buttressing is generated in first 5 km of ice downstream of the GL.

We also studied perturbations of increasing size to the thickness of the ice shelf. Here, again, we found that large changes to the geometry of the ice shelf are required to produce significant changes in GLF, with 30 m of thinning across the shelf inducing a 10% increase in GLF and over 200 m of thinning required to produce a doubling of GLF. Finally, we examined the response in ice velocities to the ungrounding of the ice shelf from the Bawden and Gipps ice rises, and found that whilst there are significant local speedups of around 50%, there was a limited instantaneous increase in GLF of 2.2%. This suggests that whilst these pinning points control the local ice-shelf dynamics, they only provide a small amount of the total buttressing of the LCIS. These diagnostic experiments have given us new insight into the total amount of buttressing provided by the LCIS and where in the ice shelf this buttressing is generated. The form of the transient response to these perturbations remains an open question to be explored in further work.

### Appendix A: L-curve analyses and $A$ and $C$ fields

As discussed in Sect. 2.3, L-curve analyses were used to determine the $\gamma_{aA}$, $\gamma_{aC}$, $\gamma_{sA}$ and $\gamma_{sC}$ parameters in Eq. 7. The values of $\gamma_{sA/C}$ were separately varied over 6 orders of magnitude and the optimisation procedure was carried out for each value of $\gamma_{sA}$ and $\gamma_{sC}$ whilst the other three parameters were held constant. This method was then repeated, separately varying $\gamma_{aA}$ and $\gamma_{aC}$ over five orders of magnitude whilst the other three parameters were held constant. The model-observation misfits for the different amounts of regularisation applied are shown in Fig. A1. The chosen values after the L-curve analyses were $\gamma_{aA}$ and $\gamma_{aC} = 1$ and $\gamma_{sA}$ and $\gamma_{sC} = 1000$. This amount of regularisation was then used to determine the $A$ and $C$ fields used throughout the experiments detailed in the main text.

The fields for the rate factor, $A$, and basal slipperiness parameter, $C$ are shown in Fig. A2a and b respectively. Examining the rate factor field, we can see that softer, more deformable ice is found in the shear margins of the ice shelf and between flow units emanating from the tributary glaciers. Higher values are also seen in regions where rifts are located (e.g. upstream of the Gipps Ice Rise, and at the location of the rift that eventually formed the A68 iceberg).

### Appendix B: Linearity of the GLF response to thinning

Fig. B1 focuses on the GLF response to smaller 'uniform' ice-shelf thickness perturbations. Over the first $\sim 100$ m of applied thinning, the response in GLF is approximately linear, with a 0.36% increase in GLF for every 1 m of ice-shelf thickness

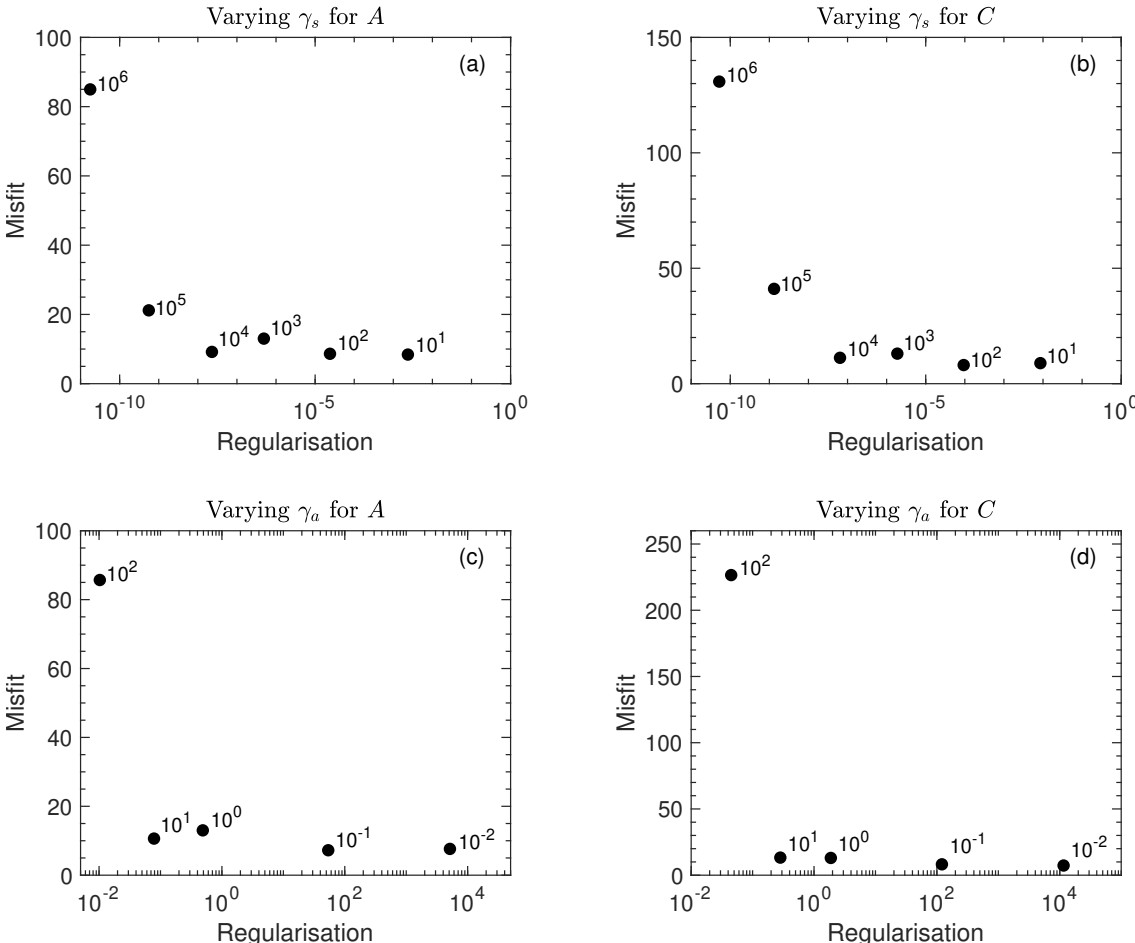

**Figure A1.** L-curves used to determine the amount of regularisation to apply in the optimisation of the $A$ and $C$ fields. **(a)** is the misfit term ($I$ in Eq. 6) plotted as a function of the regularisation applied ($R$ (from Eq. 7) divided by $\gamma_{sA}^2$), whilst varying $\gamma_{sA}$ (value indicated by label) with $\gamma_{sC} = 1000$ and $\gamma_{aA/C} = 1000$. **(b)** is the misfit term plotted as a function of $R/\gamma_{sC}^2$ whilst varying $\gamma_{sC}$, with $\gamma_{sA} = 1$ and $\gamma_{aA/C} = 1000$. **(c)** is the misfit term plotted as a function of $R/\gamma_{aA}^2$ whilst varying $\gamma_{aA}$, with $\gamma_{aC} = 1$ and $\gamma_{sA/C} = 1000$. **(d)** is the misfit term plotted as a function of $R/\gamma_{aC}^2$ whilst varying $\gamma_{aC}$, with $\gamma_{aA} = 1$ and $\gamma_{sA/C} = 1000$.

removed. For the 'proportional' perturbations (Fig. B2) there is also an initial linear regime which extends to a 10% thinning of the shelf. In this regime there is a 1.4% increase in GLF for every 1% reduction in ice-shelf thickness.

For the 'uniform' perturbations, initially the deviation from an exact linear response is below the straight line plotted through the origin and the 0.1 m thinning point. This suggests that increasing thickness perturbations produce a relatively smaller 390 increase in GLF when uniform thinning is applied. However, Fig B2 shows that when the shelf is thinned in proportion to the total ice-shelf thickness at each node, this behaviour is suppressed, and the relative response in GLF steadily increases as the proportion of the ice-shelf thickness removed increases.

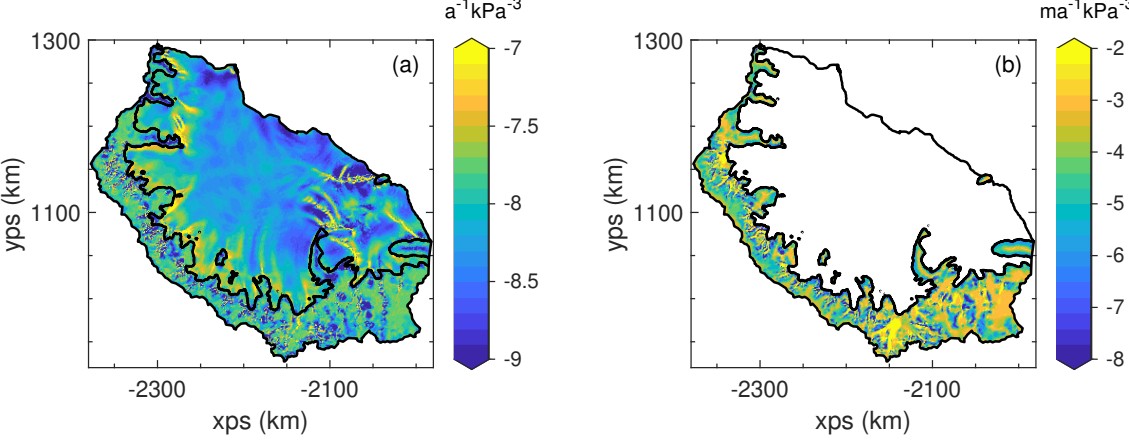

**Figure A2.** Maps of **(a)** the rate factor, $A$, in Glen's flow law (Eq. 3) and **(b)** the basal slipperiness, $C$, in the Weertman sliding law (Eq. 5) after optimisation as set out in Sect. 2.3. The colour bars have been saturated to allow the spatial detail in both parameters to be clearly seen.

The way in which the GLF response to thinning deviates below the initial linear regime is an interesting phenomena that has yet to be explained, but has been observed in previous studies on other ice shelves (e.g. Fig. S2 in Reese at al., 2018). It

suggests that when a uniform perturbation to ice-shelf thickness is applied across a shelf, over a certain range of perturbation size (here $\sim 1 - 50$ m), the relative increase in GLF is progressively reduced. From the lack of evidence of this behaviour in the proportional thinning experiments, we can see that is related to the distribution of the thickness perturbation across the shelf, and the cause of this is still unknown.

## Appendix C:  Mesh resolution dependence

To test the dependence of our results on mesh resolution, we repeated all of the experiments outlined in Sect. 2 with four additional computational meshes. In each case, the resolution around the GL was held constant at $250$ m. This was to ensure that the calving and thinning experiments conducted with each mesh had the same physical extent – as a coarser resolution at the GL would mean that the calving experiments would not penetrate as close to the GL, and ice-shelf thinning would also not be applied as close to the GL.

The four additional meshes multiplied the original mesh resolution factors (as outlined in Sect. 2 based on ice velocities, strain rates and whether or not the element was afloat) by 0.5, 2, 3 and 4 respectively. The results of 10 experiments (5 calving and 5 uniform thinning) with the 5 different meshes are shown in Fig. C1. From this we can see that the response in GLF is consistent across the 5 different mesh resolutions for each experiment, and therefore any mesh dependence of our results is negligible.

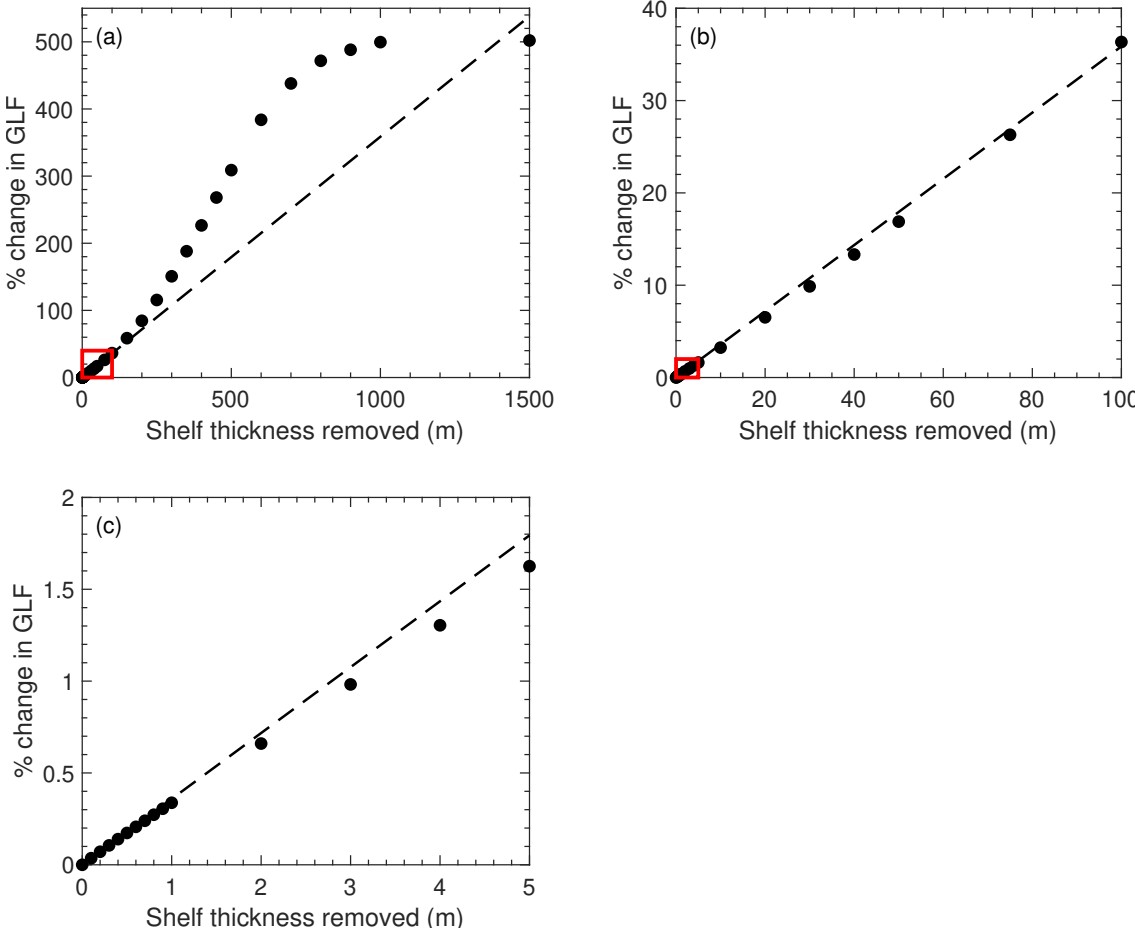

**Figure B1.** The percentage change in grounding line flux is plotted as a function of the 'uniform' thickness perturbation applied **(a)**. The dashed line is the straight line that passes through the origin and the 0.1 m 'uniform' thinning point. **(b)** is the region shown in the red box in panel (a), and **(c)** is the region in the red box in (b).

## Appendix D:  Comparing constant and variable ice-density approaches

In the results presented earlier in this study, a horizontally spatially variable ice density was used, which is show in Fig. D1. But many ice-flow models do not account for spatial gradients in ice density and instead use a constant value, typically of $917 \text{ kgm}^{-3}$. Here, we test the sensitivity of our findings to this different definition of ice density by performing the A68 iceberg calving and the idealised calving experiments with a model setup that uses a constant, depth-integrated ice density of $917 \text{ kgm}^{-3}$ across the whole computational domain.

To setup the model in this way, we again took the ice thickness from the BedMachine data set, but no longer applied the correction to the upper ice surface to account for the firn air content. Therefore, in the setup with a constant ice density of

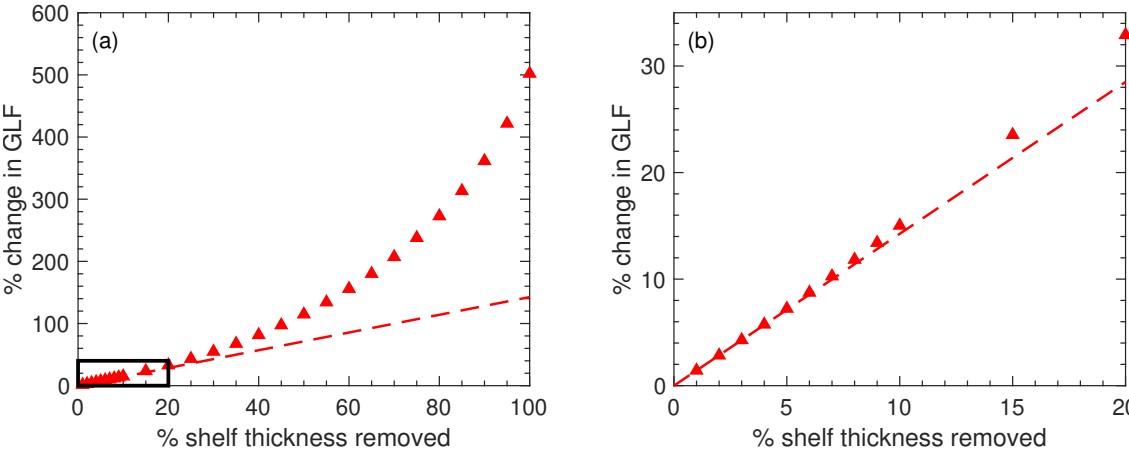

**Figure B2.** The percentage change in grounding line flux is plotted as a function of the 'proportional' thickness perturbation applied **(a)**. The dashed line is the straight line that passes through the origin and the 1% 'proportional' thinning point. **(b)** is the region shown in the black box in panel (a).

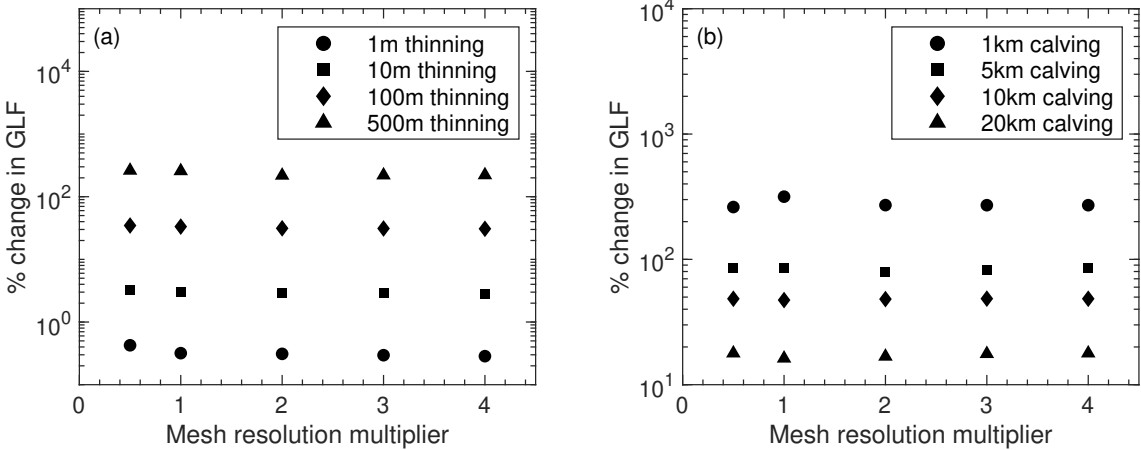

**Figure C1.** Mesh convergence analysis: Here, the percentage change in grounding line flux for five of the uniform thinning experiments **(a)** and five of the idealised calving experiments **(b)** are plotted for meshes in which the element sizes were half, double, triple and quadruple the size of the original mesh used for analysis in the main text.

$917 \ \mathrm{kgm}^{-3}$, the ice surface is lower than in our variable ice density experiments. However, the total mass of ice is the same in each case.

We then generated a new computational mesh with the same definitions of element size as outlined for the main experiments (in Section 2.2). We then performed a new optimisation procedure to generate $A$ and $C$ fields for this constant ice density setup, again using the same regularisation parameter choices as for the variable ice density setup (see Appendix A). From this new initial condition with a constant ice density, we repeated the A68 iceberg calving and idealised calving front retreat experiments

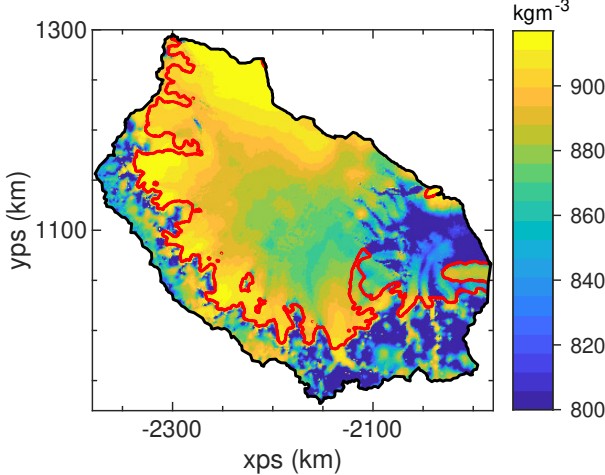

**Figure D1.** A map of the depth-integrated, spatially variable ice density (with a minimum ice density of $800 \text{ kgm}^{-3}$) as used in the main experiments in this study. The red line shows the grounding line and the black line the boundary of the model domain.

as set out in Section 2.4. The results of these experiments are shown in Fig. D2. We can see that the impact of using a constant
ice density compared with a spatially variable ice density is minimal for both small and large perturbations to they system, and
that our conclusions are not changed depending on which approach is taken. The largest differences in the modelled response
to the A68 calving between the constant and variable ice density experiments, as shown in Fig. D2b, are in the regions of the
ice shelf that have the lowest ice density in the variable setup (see Fig. D1), and therefore have the largest contrast with the
density of $917 \text{ kgm}^{-3}$ used in the constant density setup.

**Appendix E: Sensitivity to the sliding law stress exponent**

The impact of different stress exponents ($m$ values) in the Weertman sliding law (Eq. 5) on the response to perturbations in
ice-shelf buttressing was also tested. In these experiments, we used the same computational mesh as in the main experiments,
but performed new optimisation procedures to generate $A$ and $C$ fields that were consistent with the different versions of the
sliding law being used. The range of $m$ values tested was from $m = 1$ (a linear version of the Weertman law) to $m = 7$, a
more highly nonlinear version of the sliding law than the default of $m = 3$ used in our main experiments. From each of these
initial conditions, we performed some of the idealised calving experiments (moving the calving front to 1, 5, 10 and 20km
downstream of the main grounding line) and calculated the instantaneous change in GLF for each of the sliding law variations.
The results of these experiments are shown in Fig. E1.

     The pattern of instantaneous GLF increase in response to perturbations across the range of $m$ values tested is similar to that
shown in the Supplementary material of Gudmundsson et al. (2019) (Fig. S9), with an increasing GLF response to perturbations
as $m$ increases. The ice flow model Úa, used here, was also used in that study. Whilst the absolute values of the GLF response
to calving perturbations do vary depending on the value chosen for the stress exponent (due to the way in which this affects the

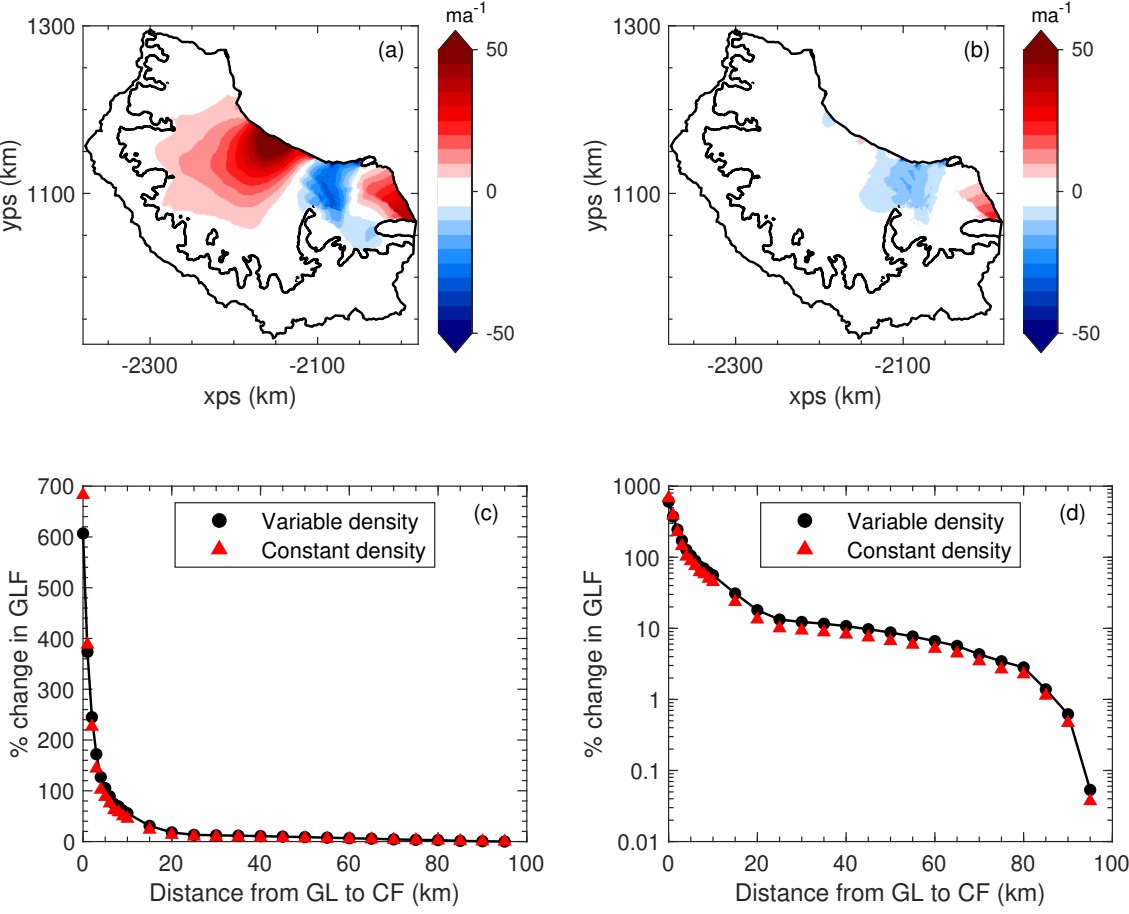

**Figure D2.** The results of calving experiments using a constant ice density of $917 \, \mathrm{kgm^{-3}}$. **(a)** shows the modelled response to the calving of the A68 iceberg using the constant density setup, whilst **(b)** shows the difference between the constant ice density response and the variable ice density response to the A68 calving. **(c)** and **(d)** show the instantaneous grounding line flux response to the idealised calving experiments for the variable ice density setup (black, the same data as in Fig. 4a and b) and the constant ice density setup (red).

stress balance at the grounding line), the finding that the vast majority of the buttressing capacity of the LCIS is generated in the first few kilometres of ice downstream of the GL remains consistent.

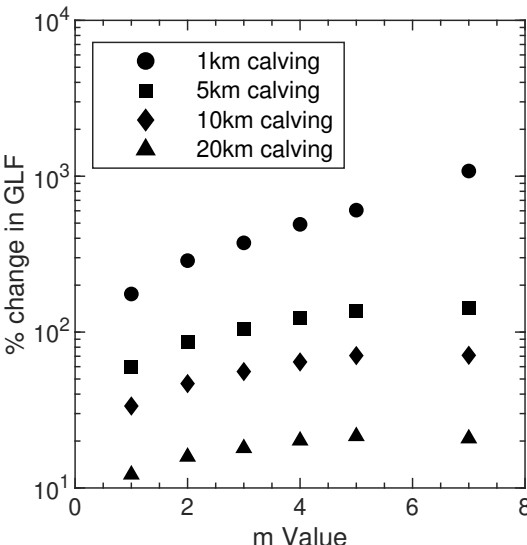

**Figure E1.** The instantaneous increase in grounding line flux for four of the idealised calving experiments, plotted as a function of the different stress exponents (m values) tested in the Weertman sliding law (Eq. 5). The four calving experiments are those in which the calving front was moved to 1, 5, 10 and 20km downstream of the main grounding line, as described in Sect. 2.4.

*Code and data availability.* The ice velocity dataset "ENVEO, Antarctic Ice Sheet monthly velocity maps from Copernicus Sentinel-1, 2014-2019, ESA Antarctic Ice Sheet CCI [v1.1]" is available on request from https://cryoportal.enveo.at. The source code for Úa is available at https://doi.org/10.5281/zenodo.3706624 (Gudmundsson, 2020). The model configuration scripts, raw model output and analysis scripts are available from the authors on request.

*Author contributions.*  All authors were involved in conceiving the study. TM conducted the modelling experiments, carried out the analysis
and wrote the manuscript. GHG and JLB discussed the analysis and provided comments on the manuscript.

*Competing interests.*  The authors declare that they have no conflict of interest.

*Acknowledgements.*  TM is supported by a GW4+ Doctoral Training Partnership studentship from the Natural Environment Research Council [NE/L002434/1]. We thank Jan Wuite at ENVEO for providing the monthly observations of Antarctic ice velocity.

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
