# Peer review of "The instantaneous impact of calving and thinning on the Larsen C Ice Shelf"

_The Cryosphere, 2021_

## Referee Comment (RC1)

[referee-annotated manuscript omitted]

---

## Referee Comment (RC2)

**General comments**

This paper studies the impact of the recent large Larsen C 's calving event (iceberg A68) on the stability of the ice shelf and the ice velocity change both on the ice shelf and at the grounding line. This first simulation is followed by a series of synthetic perturbations of the ice shelf to study the potential effect of future calving event or large thinning.

The paper is well written and easy to read, with a sufficient amount of detail (generally). The paper is in line with a series of previous papers treating the buttressing effect of ice shelves: e.g., Fürst et al. (2016), Reese et al. (2018), Gudmundsson et al. (2019). In that sense, I think that the paper proposes a limited novelty in the field.

I think that the real case modeling of the A68 calving event and the comparison of the model velocity change with Sentinel-1 SAR observations is particularly well treated and gives a valuable and additional piece of information to the current literature on buttressing and the effect of calving events. Indeed, such processes are often treated only from an observation point of view or only from a model point of view. I really enjoyed the combination of the two here, although a previous and similar work has been conducted by Borstad et al. (2017), which is "acknowledged" by the authors.

However, I have some concerns about the significance of the instantaneous response of ice shelves and tributary glaciers to downstream ice mass loss. Since such steady modeling does not account for any transient, it seems only a good tool to simulate the effect of small variations for which the transients (ice thickness evolution, advection of the ice front post calving event, "degrounding" of grounding zones due to dynamic thinning, etc.) remain limited. I fear that the results obtained for the largest perturbations applied (thinning or calving of most of the ice shelf) are very theoretical (in the sense that they do not capture the entire physics of an ice-sheet evolution) and of limited value to assess the real effect of an entire (or substantial) collapse of the ice shelf. Also, community work such as the ABUMIP experiment (Sun et al., 2020) already focuses on the transient effects of similar events (ice shelf collapse).

I therefore think that some transient simulations, allowing the ice shelf and its tributary glaciers to evolve after a calving event would greatly improve this study, bringing more insights on the real impact of calving on Larsen C but also other ice shelves. It seems that the authors intend to do such work in the future (as they raise some questions at the end of the discussion) but maybe a part of this work should be included in the present publication.

I also have a few minor comments that are listed below.

**Minor comments**

Line 45: Similar work has also been conducted by the second author of (Gudmundsson et al., 2019) for the entire Antarctica. I would cite this work in the introduction. It would be

interesting to make a few comparisons between this new study and Gudmundsson et al. (2019), to see if numbers agree.

Line 63: Could you refer to Figure 1 here, so that the reader directly looks at the map and locates the two IR.

Line 86: I'd specify that n=3 is a standard in ice modeling (rather than a choice from the authors).

Line 87: I'd delete the comma between "non-linear" and "Weertman".

Line 92: I am not well aware of the impact of the m exponent in Hill et al. (2018) but I think that Gudmundsson et al. (2019) show a substantial impact when changing m over some regions (up to 40% of GLF in their supplementary Figure S9). I agree that even such change would not affect the main conclusions of the study. However, in Gudmundsson et al. (2019), their perturbation of the ice shelf is not as extreme as the cases where your remove the most of the ice shelf or where you "thin" the ice shelf by more than 50%.  I therefore think that applying their conclusion to this paper is a little too much. A little more detail on the effect of the m exponent, with an additional experiment testing this effect, would be useful to the paper.

Line 95: "The calving front location represents a pre-July 2017 state". Could you precise how the calving front was defined (?), as it supposedly does not come from Cook and Vaughan which was published in 2010.

Line 103: It is great that the authors tested the impact of resolution. As the impact seems negligible (conclusion of Appendix C), I'd suggest to specify the negligible effect in the main text too (avoiding to jump to the Appendix during a quick read).

Line 111: What is the impact of the density on the model results? Could you also add a map of the integrated density (maybe in a supplementary material)?

Line 113: What is the impact of using one velocity dataset (MEaSUREs) or the other (Sentinel-1)? I believe that the the MEaSUREs InSAR-based Antarctic Ice Velocity v2 data set (Rignot et al., 2011; Mouginot et al., 2012) is a "post-publication" version and that the data should be cited properly too. Instructions are provided at this NSIDC link: https://nsidc.org/data/NSIDC-0484/versions/2. Could you also give a reference for the ENVEO data?

Equation 6, Line 123–127, and Appendix A:
- I understand that you use the same regularization parameter for both the A and C regularization. What is the order of magnitude of A and C (or their gradient), there must be a risk to over or under regularize one of these parameters if they have a different order of magnitude.
- Would the method beneficiate from a proper 3D treatment of the L-curve (e.g., Fürst et al., 2015) where both regularization parameters are varied together? I think that would allow a

better optimization of the model (even if your velocity misfit is relatively low). Again, considering different regularization parameters for functions of A and C could be useful.

- What are these priors? You should state these since the choice of prior can affect the optimization.

Section 2.5: I have one concern about the effect of the spatially varying ice density when instantaneously thinning the ice shelf. I expect the vertically averaged density to vary as the ice thin. Also, thinning by surface ablation/melting will have a different impact on the averaged density than basal melting. I think that since you use a spatially varying vertically integrated density, the density change with thinning should be investigated. If density has no significant effect, maybe mention that despite the model accounts for a varying density, its effect is not important to this study.

Figure 4: please specify that the red line in (c) is the grounding line.

Line 201: Please refer here to Sec. 2.5 that explains the two types of thinning (also use the same wording to define them along the paper).

Line 225: You mention that the velocity decrease is likely an artefact in the model due to the timing of the velocity you assimilated for initializing the model. I believe that the negative response could also be partially due to a change of stress state and that acceleration in a region can lead to a deceleration in another one, although such behavior might be only possible when considering a transient model.

Line 280: I would change "dynamic response" for "instantaneous response" as there is no dynamical/transient effect in this study.

Line 286: The (instantaneous) buttressing effect of the ice rises is indeed relatively small but if the ice rises had to disappear, the dynamical effect might be more important than a 2.2% change in grounding line flux. In my opinion, this is where a transient experiment would bring very valuable insights.

Section 5 (Conclusions): Following my comments on the importance of considering the dynamics of the system and not only the instantons buttressing effect, I think that some of the results, such as the minor contribution of most of the ice shelf (e.g., up to 5 km from the grounding line) should be tempered.

**References**

Fürst, J. J., Durand, G., Gillet-Chaulet, F., Merino, N., Tavard, L., Mouginot, J., Gourmelen, N., and Gagliardini, O.: Assimilation of Antarctic velocity observations provides evidence for uncharted pinning points, The Cryosphere, 9, 1427–1443, https://doi.org/10.5194/tc-9-1427-2015, 2015.

Sun S. et al. (2020). Antarctic ice sheet response to sudden and sustained ice-shelf collapse (ABUMIP). Journal of Glaciology 66(260), 891–904. https://doi.org/ 10.1017/jog.2020.67

---

## Author Comment (AC1)

We thank Referee #1 for their assessment of the study and their suggestions for improving this manuscript. Below, the reviewer comments are in black, with our response following in blue.

**General comments:**

"I feel there is a need to include some caution about these results as they are limited to instantaneous changes in grounding line flux. It would be helpful to either include additional experiments that considered the temporal evolution of the system. Or, emphasize clearly that this assessment is limited to instantaneous changes and could be considered as a sensitivity exercise rather than predictive of any future scenario."

and

"I think it is important to emphasis to the reader that these conclusions/results are based on the instantaneous response and that it is unclear how the system will evolve temporally. (This point is mentioned at the end of the discussion, but it should be more prominent throughout.) An uninitiated reader may not realize this nuance. I think it is important that this point is addressed prior to publication."

Thank you for highlighting these important points throughout your review. We have addressed these comments in our general response to both referees.

**Specific Comments:**

Title: Not sure that you can claim to be considering future calving events. But you are considering a whole range of possible scenarios (thinning and ungrounding).

Possible alternative: The impact of iceberg calving and ice-shelf thinning on the Larsen C Ice Shelf

We agree with this point, thank you. The title will be amended accordingly.

Larsen C ice shelf -> Larsen C Ice Shelf.  (Repeat this for Gipps Ice Rise and Bawden Ice Rise in text too)

Done

Abstract:

I like how you've been able to quantify the buttressing capacity of the ice shelf. Again I think it would be good to emphasis the caveat that this is instantaneous, or emphasis that this is a sensitivity test to the current configuration of the ice shelf.

Thank you, we will ensure this is emphasised throughout.

Lines 42 – 45: Both of these studies identify the need to simulate the temporal evolution of the system in order to accurately model the response to ice-shelf change.

That is correct, and we would also require transient experiments to simulate the mass redistribution in response to these experiments. Here, we are concerned only with the changes in buttressing (which are instantaneous) in response to perturbations in ice-shelf geometry.

Line 107: "The surface elevation was adjusted at a few points to ensure that at least 1 m of ice was present across the whole computational domain." Is this only relevant for areas of exposed bedrock (i.e. nunataks)?

Yes, we will add a statement to that effect in the text.

Lines 109 – 111: What impact does accounting for firn density have on the results? Does the minimum ice density refer to the surface firn density?

Thank you for highlighting this point, it was also raised by Referee #2. We will now include details on the sensitivity to ice density in the appendix. The minimum ice density refers to the depth integrated ice density, and we will clarify this in the text.

Lines 113 - 114: You use MEaSUREs velocities for model initialization, but Sentinel-1 SAR for ice velocity pre- and post- calving: how do these velocities compare? Could you use Sentinal for initialization?.

The Sentinel-1 derived ice velocities are slightly higher across the ice shelf when compared to the MEaSUREs data set (the difference is on the order of 10-20 m/a). The reason that the Sentinel-1 velocity data was not used for the model initialisation is that the ice velocity in the A68 region of the shelf is much higher than in the MEaSUREs data set (on the order of 100-200 m/a) during the 2014-2016 period before the final calving event. We did not want to initialise the model velocities in the A68 region of the shelf to these higher values, which are the expression of the iceberg beginning to detach from the rest of the shelf.

Equation (6): Are you considering the difference in speed or velocity components here?

 We are considering the difference in ice velocity components. We will clarify this in the text.

Equation (7) first term on RHS: Do you mean that you constrain the gradient in the difference between A and \hat{A}? Or just the gradient in A? (which I think is the usual approach).

We constrain the gradient in the difference between A and \hat{A}, which is the same as constraining the gradient in A in our case, as we assume a spatially uniform A prior.

Line 126: How are prior estimates \hat{A} and \hat{C} chosen?

Both A and C priors are both chosen to be spatially uniform across the model domain. We will include their values in the main text and the explanation of how they are calculated. The A prior corresponds to ice at a temperature of -10 degC as suggested in Morland and Smith (1983). The C prior is derived from the sliding law, assuming a basal velocity of 100 m/a and a basal shear stress of 80kPa.

Line 127: Is it appropriate to use the same regularization parameters for A and C?

We determined that is appropriate to use the same regularisation parameters for both A and C but realise that this was not justified in the text. We will include further L-curves in the appendix that show the impact of varying the regularisation parameters for A whilst holding those for C constant and vice-versa.

Section 2.4 Calving experiments – good explanation of procedure.

Line 152-153: "the ice shelf became afloat without changing the ice thickness" Does this result in areas of thicker ice (that were formerly part of the ice rise) within the ice shelf that might induce flow "backwards" i.e. upstream relative to the large-scale flow?

This does occur at the Gipps ice rise, but not at the Bawden ice rise. At the Gipps ice rise, the ice thickness does increase as you move downstream across what was the grounded ice rise. However, the gradient is smaller than in other regions across the ice shelf, such as where crevasses cut across

the flow, and this increase in ice thickness occurs across one or two model nodes. There is not such a change in the stresses as to induce 'backwards' flow.

Section 3 Results – I think it would be good to include an initial sentence here that states that this section is split into subsections reporting the results of each of the scenarios outlined in the methods.

Done

Line 176: "no observable, transient response to the calving of the A68" – it would be good to add that from the dataset spanning 5 years there has been not appreciable change in velocity and no long-term response to calving.

Done

Lines 185 – 188: "Therefore, Fig. 4a and b show what proportion of the total buttressing is provided by each section of the ice shelf removed in the series of calving perturbations. From this, we see that over 95% of the total buttressing is provided by ice in the first 25 km downstream of the GL, and that over 80% is generated in the first 5 km of ice immediately downstream of the GL." Statements like these are where I feel there needs to be some further note that this is just based on the instantaneous response. This could be framed in terms of a sensitivity, or the contribution to buttressing made by the current configuration of the ice shelf.

As stated in our general response to both referees, we will clarify that these are instantaneous calculations about the current configuration of the shelf. But we do think that the passage that you have quoted is correct, in that we have calculated the total buttressing provided by the shelf, and then found by how much that buttressing is reduced by moving the calving front to different locations.

Section 3.3 Ungrounding experiments: Again these results should be treated with caution. Studies such as Favier & Pattyn 2015 have demonstrated the influence of an ice rise on the temporal evolution of grounding-line flux and position.

Thank you for highlighting that study. We will emphasise that these experiments only investigate the buttressing provided by the ice rises, rather than the impact of their removal on the transient mass redistribution.

Line 202: "over 200 m of thinning is required to produce a doubling of GLF" – how does this compare in terms of areal extent to the idealized calving experiments? A doubling in GLF seems huge, how does this translate into contributions to sea level rise? I expect it would be more than a doubling in SLR contribution.

Thank you for this suggestion. We will include a line in the text that compares the ice-shelf mass removed in the calving experiments to the thinning experiments. However, we have chosen not to frame these results in terms of a SLR contribution, as transient experiments would be needed to determine the mass redistribution in response to these perturbations.

Lines 203 – 204: "The maximum ice thickness in the shelf was 1,400 m, so a thinning perturbation larger than this had to be applied to reduce the ice shelf to the minimum thickness of 1 m everywhere." I think there must be a typo here – thinning of 1,400m would lead to no ice shelf remaining?

Thank you for highlighting this, we agree that it is poorly worded and will correct this. There is an algorithm in the model code that ensures the ice thickness is never less than 1m, so by thinning the shelf by 1,500m the ice shelf is effectively set to the minimum ice thickness everywhere. But really the maximum amount of thinning applied was 1,489m the maximum value of ice-shelf thickness at any of the ice-shelf nodes.

Lines 228 – 231: "Therefore, the dynamic response to the detaching of the nascent A68 iceberg will have already taken place in this region, and this response is included in the ice velocity data used to initialise our model. Finally, in the model we essentially force the already detaching iceberg to have contact with ice upstream, inducing an artificial 'pulling' effect on this upstream ice, which is removed when the iceberg is calved from the domain." This is evident in Fig 2C where the model assimulation produces speeds less than observations.

Thank you for raising this point, we will refer the reader to Fig 2c at this point.

Lines 238 – 242: Again important to emphasis that this assessment is based on instantaneous response.

Agreed, we will address this throughout the manuscript.

Lines 246 – 247: "We argue that it is this second definition, the integrated impact of changes in ice-shelf geometry on stresses at the GL and consequently on GLF, that is the key measure of the buttressing capability of ice shelves." The buttressing capacity of the ice shelf in its current configuration. It is noted by Furst et al., 2016 that after a calving event occurs the stress field within the shelf will evolve, such that the initial assessment is no longer valid.

We will clarify that it is changes to the current configuration that we are addressing, and that questions around transient mass redistribution in response to a perturbation are not presented here.

Lines 253 – 254: "by removing the entire ice shelf and calculating the instantaneous response in GLF we are able to quantify the total amount of buttressing that the LCIS provides" Good point!

Thank you.

Lines 254 – 256: Again important to emphasis that this based on instantaneous response.

Done.

Lines 269 – 274: This seems like an artifact of the modelling rather than something that would occur in the real world - surely the 1m (and even more so with 0.001m) thick ice shelf would break or buckle when trying to resist the flow of 1000+m ice streams. I think it is important to acknowledge this.

Yes, we will clarify that this was just a technical exercise to ensure that the chosen minimum ice shelf thickness (which is used for numerical reasons) did not unduly impact the results of the experiment.

Lines 275 – 279: How does this magnitude of basal melting compare to surface accumulation? By how much is the ice shelf actually thinning?

As these experiments are instantaneous, there is no surface accumulation field applied to the model. Therefore, the amount of thinning stated is the actual amount by which the ice-shelf thickness is reduced.

Lines 286 – 288: "This suggests that, whilst these two ice rises may exert a significant control on the flow of the shelf upstream of the pinning points, they do not exert a strong mechanical control on the ice flux at the GL, and only contribute a small amount to the total buttressing capability of the shelf." This statement is again based on the instantaneous response. In a temporal sense removing the ice rise would have a massive effect on the flow of the shelf, changing its geometry and thickness, and later impacting the flow at the grounding line.

We are happy to reiterate that the results presented here are instantaneous and that transient evolution is not explored. But from our early transient experiments, we find that on losing basal contact at the ice rises the instantaneous change in GLF is maximal, and that the GLF decays towards its initial value through time.

Lines 293 – 299: This is a very important point, which should be highlighted much earlier.

It is good that you have acknowledged this point, but I think you need to explore this point more or at least note that these results are not complete.

Someone reading this may easier assume that the large parts of the shelf can be removed without increasing discharge, which is not the case! Instantaneously maybe, but once the geometry/thickness of the shelf has adjusted to the imposed change the GLF will change too. You have only explored the buttressing generated by the current configuration of the LCIS.

We hope that the new manuscript will be clear on this point throughout. But as stated in our general response to both reviewers, we do want to emphasise that the buttressing effect is necessarily 'instantaneous' as it relates to the stress state in the ice. We think that our statements about which parts of the ice shelf generate the buttressing are correct, but that does not necessarily relate to the transient evolution. Again, our initial transient experiments suggest that the instantaneous change in GLF is maximal, and it decays towards its initial value, as the tributaries thin in response to the acceleration.

Lines 311 – 313: "Here, again, we found that large changes to the geometry of the ice shelf are required to produce significant changes in GLF, with 30 m of thinning across the shelf inducing a 10% increase in GLF and over 200 m of thinning required to produce a doubling of GLF." It may be good to put these changes in GLF into context in terms of the total mass balance of the catchment, i.e. total accumulation - GLF. What does a 10% increase in GLF mean for sea-level contributions?

We can certainly include a further statement here on the mass balance of the catchment. The reference GLF is stated earlier in the manuscript, but we can restate that here as well. As stated in a response above, we do not use a surface mass balance field in these instantaneous experiments, and do not want to discuss the results in terms of SLR contribution, as transient experiments would be needed to do so.

Lines 315 – 317: "This suggests that whilst these pinning points control the local ice-shelf dynamics, they only provide a small amount of the total buttressing of the LCIS." I don't think this is a good sentence to end on. The statement is based on the instantaneous response. And it's probably not the most significant finding from the study.

Thank you for highlighting this, we agree that it is not the most significant finding from the study and therefore should not be the final sentence of the text. But we do think that it is correct to say that the ice rises only provide a small amount of the total buttressing of the LCIS in its current configuration; that is what our experiments are designed to elicit. We agree though that this should

not be interpreted as a comment on the potential transient mass redistribution following a perturbation.

Appendix B: Linearity of GLF response to thinning: It would be good to compare this with theoretical results such as Pegler 2018 and Haseloff & Sergienko 2018 as the buttressing force in these cases features an integral along the length of the ice shelf with an integrand containing the vertical integrated (i.e. thickness) longitudinal stress.

Thank you for highlighting this work, we will investigate the links to these theoretical results, and include that in this appendix.

References:

Favier, L., & Pattyn, F. (2015). Antarctic ice rise formation, evolution, and stability. Geophysical Research Letters, 42(11), 4456–4463. https://doi.org/10.1002/2015GL064195

Haseloff, M., & Sergienko, O. V. (2018). The effect of buttressing on grounding line dynamics. Journal of Glaciology, 64(245), 417–431. https://doi.org/10.1017/jog.2018.30

Pegler, S. S. (2018). Marine ice sheet dynamics: The impacts of ice-shelf buttressing. Journal of Fluid Mechanics (Vol. 857). https://doi.org/10.1017/jfm.2018.741

---

## Author Comment (AC2)

We thank Referee #2 for their review and their suggestions for improving this manuscript. Below, the reviewer comments are in black, with our response following in blue.

**General comments**

The paper is well written and easy to read, with a sufficient amount of detail (generally). The paper is in line with a series of previous papers treating the buttressing effect of ice shelves: e.g., Fürst et al. (2016), Reese et al. (2018), Gudmundsson et al. (2019). In that sense, I think that the paper proposes a limited novelty in the field.

The key advances over the previous work mentioned are: The replication of the A68 iceberg calving event, the calculation of its impact on the upstream ice (including across the grounding line) and a comparison with observations; and secondly, the calculation made of the total amount of buttressing provided by the LCIS and then the exploration of what proportion of this total is generated in different regions and by different features (e.g. ice rises) within the shelf.

I think that the real case modeling of the A68 calving event and the comparison of the model velocity change with Sentinel-1 SAR observations is particularly well treated and gives a valuable and additional piece of information to the current literature on buttressing and the effect of calving events. Indeed, such processes are often treated only from an observation point of view or only from a model point of view. I really enjoyed the combination of the two here, although a previous and similar work has been conducted by Borstad et al. (2017), which is "acknowledged" by the authors.

Thank you for this comment, we agree that similar experiments were conducted by Borstad et al. and that was the reason we compared our results. In our work we extended the model domain to include the grounded tributary glaciers as well (Borstad et al. modelled the ice shelf only), allowing us to explicitly assess the impact on GLF.

However, I have some concerns about the significance of the instantaneous response of ice shelves and tributary glaciers to downstream ice mass loss. Since such steady modeling does not account for any transient, it seems only a good tool to simulate the effect of small variations for which the transients (ice thickness evolution, advection of the ice front post calving event, "degrounding" of grounding zones due to dynamic thinning, etc.) remain limited. I fear that the results obtained for the largest perturbations applied (thinning or calving of most of the ice shelf) are very theoretical (in the sense that they do not capture the entire physics of an ice-sheet evolution) and of limited value to assess the real effect of an entire (or substantial) collapse of the ice shelf. Also, community work such as the ABUMIP experiment (Sun et al., 2020) already focuses on the transient effects of similar events (ice shelf collapse).

We are certainly happy to emphasise more clearly throughout the text that we are investigating instantaneous responses and buttressing in this work. This was a point that was also raised by Referee #1 and we have provided a justification and explanation for our approach in our response to general comments from both referees.
The ABUMIP paper does explore the transient mass redistribution in response to complete ice-shelf collapse, but does not examine which parts of the ice shelves are generating the buttressing. That is the question that we set out to answer in this work. The papers by Reese and Gudmundsson mentioned above examine the GLF response to perturbations in ice thickness, but do not attempt to calculate the total buttressing provided by the ice shelves and then apportion that total to different regions (and features such as ice rises) within the shelf, which we are able to do with our methodology.

I therefore think that some transient simulations, allowing the ice shelf and its tributary glaciers to evolve after a calving event would greatly improve this study, bringing more insights on the real impact of calving on Larsen C but also other ice shelves. It seems that the authors intend to do such work in the future (as they raise some questions at the end of the discussion) but maybe a part of this work should be included in the present publication.

We completely agree that transient simulations of these perturbation experiments are interesting and important, and we are currently undertaking this work. We have set out our reasons for not including them in this study in our response to general comments from both referees.

**Minor comments**

Line 45: Similar work has also been conducted by the second author of (Gudmundsson et al., 2019) for the entire Antarctica. I would cite this work in the introduction. It would be interesting to make a few comparisons between this new study and Gudmundsson et al. (2019), to see if numbers agree.

Thank you for this suggestion, we will certainly reference this paper in the introduction and will also compare our most comparable thinning experiment results to those in the Gudmundsson paper.

Line 63: Could you refer to Figure 1 here, so that the reader directly looks at the map and locates the two IR.

Done

Line 86: I'd specify that n=3 is a standard in ice modeling (rather than a choice from the authors).

Done

Line 87: I'd delete the comma between "non-linear" and "Weertman".

Done

Line 92: I am not well aware of the impact of the m exponent in Hill et al. (2018) but I think that Gudmundsson et al. (2019) show a substantial impact when changing m over some regions (up to 40% of GLF in their supplementary Figure S9). I agree that even such change would not affect the main conclusions of the study. However, in Gudmundsson et al. (2019), their perturbation of the ice shelf is not as extreme as the cases where your remove the most of the ice shelf or where you "thin" the ice shelf by more than 50%. I therefore think that applying their conclusion to this paper is a little too much. A little more detail on the effect of the m exponent, with an additional experiment testing this effect, would be useful to the paper.

Thank you for highlighting this, we agree that our justification is lacking, and will include a further figure and discussion of the impact of using different m values in the appendix.

Line 95: "The calving front location represents a pre-July 2017 state". Could you precise how the calving front was defined (?), as it supposedly does not come from Cook and Vaughan which was published in 2010.

We defined the initial calving front by taking the ice extent of the BedMachine product (effectively the zero ice-thickness contour of the data). We will clarify this in the methods section of the manuscript.

Line 103: It is great that the authors tested the impact of resolution. As the impact seems negligible (conclusion of Appendix C), I'd suggest to specify the negligible effect in the main text too (avoiding to jump to the Appendix during a quick read).

Thank you for highlighting this, we will add a statement to that effect in the main text.

Line 111: What is the impact of the density on the model results? Could you also add a map of the integrated density (maybe in a supplementary material)?

The impact of using different depth averaged densities is small, but we will include a further figure and discussion of this in the appendix, as we agree that it would be useful to compare results using a spatially variable density to a those with a constant density of 917 kg/m^3 as is commonly used in modelling studies. We will also include a map of depth-averaged densities in this additional appendix.

Line 113: What is the impact of using one velocity dataset (MEaSUREs) or the other (Sentinel-1)? I believe that the the MEaSUREs InSAR-based Antarctic Ice Velocity v2 data set (Rignot et al., 2011; Mouginot et al., 2012) is a "post-publication" version and that the data should be cited properly too. Instructions are provided at this NSIDC link: https://nsidc.org/data/NSIDC-0484/versions/2. Could you also give a reference for the ENVEO data?

The Sentinel-1 derived ice velocities are slightly higher across the ice shelf when compared to the MEaSUREs data set (the difference is on the order of 10-20 m/a). The reason that the Sentinel-1 velocity data was not used for the model initialisation is that the ice velocity in the A68 region of the shelf is much higher than in the MEaSUREs data set (on the order of 100-200 m/a) during the 2014-2016 period before the final calving event. We did not want to initialise the model velocities in the A68 region of the shelf to these higher values, which are the expression of the iceberg beginning to detach from the rest of the shelf.

Thank you for pointing out the incorrect referencing of the MEaSUREs data, we will correct that. Unfortunately, there is no reference available for the ENVEO data, as the data has not been made publicly available online. The data is available on request from ENVEO, which we have stated in the Data Availability section.

Equation 6, Line 123–127, and Appendix A:
• I understand that you use the same regularization parameter for both the A and C regularization. What is the order of magnitude of A and C (or their gradient), there must be a risk to over or under regularize one of these parameters if they have a different order of magnitude.
• Would the method beneficiate from a proper 3D treatment of the L-curve (e.g., Fürst et al., 2015) where both regularization parameters are varied together? I think that would allow a better optimization of the model (even if your velocity misfit is relatively low). Again, considering different regularization parameters for functions of A and C could be useful.

Thank you for highlighting this, we realise we did not justify our use of the same regularisation parameter values for A and C. We will remedy this by including further figures and discussion in Appendix A to show the impact of varying the regularisation parameters for A and C separately.

• What are these priors? You should state these since the choice of prior can affect the optimization.

Thank you for pointing this out, we should have stated what the prior values are. We will include the values and a description of their calculation in the relevant passage of the methods section. Both A and C priors are chosen to be spatially uniform across the model domain. The A prior corresponds to ice at a temperature of -10 degC, as suggested in Morland and Smith (1983). The C prior is derived from the sliding law, assuming a basal velocity of 100 m/a and a basal shear stress of 80kPa.

Section 2.5: I have one concern about the effect of the spatially varying ice density when instantaneously thinning the ice shelf. I expect the vertically averaged density to vary as the ice thin. Also, thinning by surface ablation/melting will have a different impact on the averaged density than basal melting. I think that since you use a spatially varying vertically integrated density, the density change with thinning should be investigated. If density has no significant effect, maybe mention that despite the model accounts for a varying density, its effect is not important to this study.

Thank you for this suggestion. We do not account for changes in the depth integrated density when thinning the ice shelf in these experiments. The additional figures and discussion on the sensitivity to ice density mentioned in an earlier comment will address this.

Figure 4: please specify that the red line in (c) is the grounding line.

Done

Line 201: Please refer here to Sec. 2.5 that explains the two types of thinning (also use the same wording to define them along the paper).

Done, and thank you, we will ensure that the same terminology is used throughout.

Line 225: You mention that the velocity decrease is likely an artefact in the model due to the timing of the velocity you assimilated for initializing the model. I believe that the negative response could also be partially due to a change of stress state and that acceleration in a region can lead to a deceleration in another one, although such behavior might be only possible when considering a transient model.

Thank you for this suggestion. Yes, we agree that this response is possible, but would be hard to disentangle from the potential artefacts of the model initialisation approach in this region.

Line 280: I would change "dynamic response" for "instantaneous response" as there is no dynamical/transient effect in this study.

Done

Line 286: The (instantaneous) buttressing effect of the ice rises is indeed relatively small but if the ice rises had to disappear, the dynamical effect might be more important than a 2.2% change in grounding line flux. In my opinion, this is where a transient experiment would bring very valuable insights.

We agree that the transient mass redistribution in response to the loss of the ice rises is a very interesting question, and we are currently doing these experiments. But we think that this is still an important result, and that the buttressing provided by the ice rises can be fully captured by these instantaneous experiments. From our initial transient experiments, we find that this 2.2% increase in GLF is a maximal response to the perturbation, and that the GLF decreases towards its initial value through time, due to the thinning of tributaries in response to their initial acceleration. However, our reasons for not including these in the present work are set out in our response to general comments from both referees.

Section 5 (Conclusions): Following my comments on the importance of considering the dynamics of the system and not only the instantons buttressing effect, I think that some of the results, such as the minor contribution of most of the ice shelf (e.g., up to 5 km from the grounding line) should be tempered.

We are happy to clarify the language throughout the manuscript to ensure that the reader knows that these are instantaneous experiments and responses. But we think that ice-shelf buttressing can be fully characterised by these instantaneous experiments. The transient mass redistribution in response to changes in buttressing is a crucial, but separate question to the one we are trying to answer in this manuscript.

**References**
Fürst, J. J., Durand, G., Gillet-Chaulet, F., Merino, N., Tavard, L., Mouginot, J., Gourmelen, N., and Gagliardini, O.: Assimilation of Antarctic velocity observations provides evidence for uncharted pinning points, The Cryosphere, 9, 1427–1443, https://doi.org/10.5194/tc-9-1427-2015, 2015.
Sun S. et al. (2020). Antarctic ice sheet response to sudden and sustained ice-shelf collapse (ABUMIP). Journal of Glaciology 66(260), 891–904. https://doi.org/ 10.1017/jog.2020.67

---

## Author Response (AR1)

Dear Tom Mitcham and co-authors,

I want to sincerely thank you and your co-authors for having thoroughly addressed most comments raised by the reviewers. Unfortunately, you did not attach a track changed version of your revised manuscript (I might have missed it; the lats file I can see is from April 14). So I base my judgement on the answers you directly gave to the review comments.

Thank you. In this submission we have attached a revised version of the manuscript addressing the comments of the two reviewers as well as a track changes version.

In my opinion, the most important request was on additional transient perturbation experiments to distinguish your work from predecessor studies. In the general answer to both reviewers you explain that such transient experiments require a substantial revision and expansion of the article draft. To some extent, I can understand this argument. Yet as this was a major comment from both reviewers, I suggest that your article enters a second review round to inquire their perspectives.

Thank you, we hope that we have addressed these concerns in the revised manuscript being submitted for a second round of reviews.

When uploading the revised article draft, also keep an eye out for the option to upload a manuscript version showing track changes. This will certainly facilitate the work of the reviewers. If this was no longer possible at this stage, get in touch with the editorial office to decide on the best option to make a track-changed document available.

We have now uploaded a track changes version of the revised manuscript as well.

Best,

Johannes Fürst

---

## Referee Report (RR1)

**General comment**

I want to thank the authors for having addressed most of my comments during their revisions. The current paper is nicely written and is well detailed. I also appreciate the new simulations they conducted in the Appendix to better emphasis some arguments from the main text.

Yet, my main concern (as for my first review) is the significance of the instantaneous response of ice flow to a mass loss and the lack of a proper treatment of the transient effects following such event. I still think that this drawback limits the interpretation of the experiments, especially in the case of large ice-shelf thinning experiments that should not be treated as an instantaneous event (i.e. large thinning would most likely occur over a long-time scale, giving time to the upstream flow and geometry to adapt).

As the authors argue in their response to reviewers, the aim of this work is to identify the principal buttressing points of Larsen C Ice Shelf and their impact on grounding line flux when this buttressing is released — and, in this regard, the paper is very nicely wrapped. I also understand that the authors are currently conducting transient simulations and that such work can take time. However, I think that the current paper brings only limited insight in comparison to published literature, e.g. Furst et al. (2016) Reese et al. (2018), Gudmundsson et al. (2019) and Zhang et al. (2020) for buttressing, and e.g. Borstad et al. (2017) for the calving of A68.

The main conclusion of the paper, except the quantification of the real calving event of A68, which nicely couples modeling and observations, is that only a small portion of the ice shelf (close to the GL) really matters to buttressing. While this conclusion is in line with pre-cited studies, it does not really bring new insights about buttressing and the methodology is very similar to Reese et al. (2018) and Gudmundsson et al. (2019).

In the current study, the effect of the ice rise seems much smaller than in previous studies. For example, Zhang et al. (2020) also addressed the same question with an adjoint sensitivity (more reproductible and computationally much cheaper, in my opinion) in the context of ocean-induced melting on LCIS. Their results (particularly Fig. 12a) generally agree with Furst et al. (2016). In this regard, the current study results are more in line with Borstad et al. (2013) that see only a small effect of the ice-rise loss on the GLF. However, they observe a 25% change in velocity on the shelf, that, in a transient model, could have important effect. If the authors do not go with transient simulations, I think that they should better relate their paper to the previous literature (comparing results, etc).

For all these reasons, I had a really hard time deciding what recommendation to give for the paper. To me, the lack of transient experiment and the lack of novelty with respect to the published literature are a bit redhibitory. I therefore really recommend to the authors to pursue their effort towards transient simulations and include them in this paper.

**Minor comments**

**Title:** I would change "ice-shelf thinning" for "thinning" only to avoid a repetition.

**Line 48- 63:** I think that I think that the ABUMIP paper from Sun et al. (2020) should be cited in the context of this study.

**Line 55:** I would change the sentence "Gudmundsson et al. (2019) modelled the impact of an instantaneous perturbation to Antarctic ice-shelf thickness, the spatial pattern and amplitude of which was taken from observations" for "Gudmundsson et al. (2019) modelled the impact of an instantaneous thinning of Antarctic ice-shelf on the grounded ice and GLF, with a pattern and amplitude derived from observations".

**Line 58:** change "for the last 18 years" for "from 1994 to 2017". I would also reorder the sentence "[...] the instantaneous ice velocity response due to the observed ice-shelf thinning of the last 18 years, and the subsequent reduction in buttressing" as follows "[...] the instantaneous ice velocity response and the reduction in buttressing due to the cumulated observed ice-shelf thinning from 1994 to 2017".

**Line 64 to 66:** I thank the authors for clearly stating the "instantaneousity" of their experiments. I think that these two sentences could be reshaped in only one, more powerful sentence. For example, "a series of diagnostic perturbation experiments of increasing magnitude" is a bit vague and only gets clearer reading the second sentence. The literature they used to build their experiment is also not only related to LCIS (but the sentence seems to say the opposite, i.e. "existing literature on the ice dynamics of the LCIS").

Line 70: correct the typo: "alnd"

**Line 101:** Thank you for the addition of Appendix E. Should this be Appendix A, as it is the first you reference in the main text? I do not know what are the referencing rules for Appendix.

Line 128: reformat the references in only one parenthesis?

**Equation 7 and Appendix A:** Why do you choose  $\gamma_{s_{A/C}} = 10^3$ ? I agree that your modeled velocities nicely fit observations so that might be a bit picky but the L-curve in Fig A1. (a,b) seems to show that  $\gamma_{s_{A/C}} = 10^4$  is a better value, *i.e.* with the smallest velocity misfit. Similarly, I would be tempted to say that  $\gamma_{a_{A/C}} = 10$  is a better value than 1. It also seems that you treat your L-curves independently but I assume that the choice of one parameter impacts the choice of the others, why not going with a multi-dimension L-curve like in Furst et al. (2016).

Line 142: penalize deviations?

**Line 193:** Maybe precise why did you see such negative velocity change. Reorganization of the ice flow due to the change of geometry and buttressing?

Line 237: add a comma to "[...] (of 1,500 m), the ice shelf [...]".

Line 238-239: Precise that these metrics are before perturbation.

**Line 320:** space between "m" and " $a^{-1}$ ".

**Line 320-324:** I think that this is where the instantaneous approach really shows its limit. I don't think that these numbers are really meaningful. When making estimations of future state of the ice sheet, we really want to know what will be the total mass loss, which is not possible with the approach of this paper.

---

## Referee Report (RR2)

**Review of Mitcham et al. "The impact of calving and ice-shelf thinning on the Larsen C Ice Shelf"**

November 11, 2021

**General comments:**

The manuscript by Mitcham et al. presents a very thorough and detailed numerical study of the instantaneous effect to a number of idealised ice-shelf thinning and calving perturbations to Larsen C Ice Shelf. For the calving perturbation, they find that most of the buttressing is exerted by floating ice within 5 km downstream of the present-day grounding line position. For the ice-shelf thinning experiments, the authors show that a significant thinning (ca. 200 m) is necessary to get a doubling of ice flux across the grounding line. Overall, I find the manuscript very well written and easy to follow. The provided Figures are appropriate and of high quality. The main criticism in the first round of reviews was about the novelty of the study, as a number of previous studies exist that have investigated the instantaneous response of Larsen C Ice Shelf already, albeit with slightly different foci. I should mention that I was not a reviewer in the first round of reviews.

My opinion is that the depth of experiments including additional sensitivity simulations are just enough to warrant publication as a full research article in TC, without having to undertake transient perturbation simulations. However, I think the fact that this is the instantaneous response should be highlighted throughout. To be fair the authors already do acknowledge this in several places throughout the manuscript. I think that drawing any conclusions about the future should be avoided in the manuscript (e.g. L313-317). In the following I outline my list of minor suggestions below and hope the authors find my comments helpful.

**Specific comments:**

- I think the title should already convey the information that this paper is looking at the instantaneous response. Since the paper is also focusing on ice flux across the grounding line, my suggestion would be to also include this in the title. So maybe something along

the lines of : "The instantaneous response of Larsen C Ice Shelf grounding line flux to calving and ice-shelf thinning perturbations."

- Just a comment to line 33-35. It reads like there are no studies on the transient upstream response to changes in Larsen C ice shelf thickness and extent. I would just like to point out that at least for the extreme scenario of complete ice shelf removal, we published a paper in TC in 2018 (Schannwell et al. 2018) about this. We basically find that the sea-level contribution and upstream thinning decay very rapidly (<50 years) after the perturbation.

- In section 2.1., can you please mention if you assume isothermal ice and if so what temperature and why?

- In section 2.2 L107-108, this reads like Gmsh is creating linear finite elements. As far as I know, Gmsh simply creates the triangulation or mesh, but you can use any finite element type on this mesh. Does this mean that you are using linear Lagrange elements? Would be nice to spell this out more explicitly.

- L305-307, I find the result that a 1 m or even 0.001 m thick ice shelf vs. no ice shelf gives a difference of ∼100% in ice flux across the grounding line surprising. This warrants a discussion why there is such a large discrepancy.

- L313-316 I find this scenario to be a bit too far fetched and recommend deleting this paragraph.

- Appendix E: I do not know if Ua has pressure-limited sliding laws implemented (Tsai, Budd, or Schoof sliding relation), but I think as a community we are moving towards pressure-limited sliding relations, so it would be interesting to see if the effect between these two relations is already large in the instantaneous response or only in transient simulations.

- Appendix E: Why did you rerun the inversion for different sliding law exponents? In theory you should be able to use one inversion for all different exponents because they must satisfy $C_2|u|^{m2} = C_1|u|^{m1}$, where $m1 = 1$ and $m2 = 2$ for example. You could then rearrange that for $C_2$. Maybe you can comment on that?

**Technical corrections:**

L69 outlined and labelled

L86 I think this should be reworded. The stress balance equation is always solved diagnostically. There is no time-dependence in Eq. 1. Only when you couple the stress

equation to a transport equation (e.g. ice-thickness evolution equation) does it become transient (time dependent).

L430 Whilst

**Figures:**

Fig. 1: Does the ps in the axes label stand for polar stereographic? I would consider scratching this.

Fig. 5: Is there really only speed-up in panels a-c?

Fig. C1: sizes were half

Fig. E1: (Eq.5).

Sincerely,
Clemens Schannwell

**References**

Schannwell, C., Cornford, S., Pollard, D., and Barrand, N. E.: Dynamic response of Antarctic Peninsula Ice Sheet to potential collapse of Larsen C and George VI ice shelves, The Cryosphere, 12, 2307–2326, https://doi.org/10.5194/tc-12-2307-2018, 2018.

---

## Author Response (AR2)

We thank the three reviewers and the editor for their suggestions for further improvements to the manuscript. Below, the reviewer comments are presented in black and our responses are in blue.

**Anonymous reviewer #1**

I thoroughly enjoyed reading this revised version of the manuscript. You have addressed my previous review comments and skillfully incorporated an explanation of the significant of the instantaneous response as opposed to a transient simulations.

Thank you for your time reviewing the new manuscript and your assessment of the changes we have made.

For final formatting it would be good to place Figures 3-6 earlier in the text so that it is easier to compare them with the corresponding results sections.

Thank you for this suggestion, we will make sure that this is the case in the final production version.

I spotted a few minor typos. Otherwise I recommend the manuscript is accepted.

Typos:

line 69: (outlined "alnd" labelled in Fig. 1)

Done

line 99: math format/italic (m)

Done

Lines 305-312: I think you need to end this paragraph by saying that 1m and 0.001m ice shelf would in reality provide very little buttressing and as such the difference between the thinning and calving experiments must be a result of the numerical method.

Thank you, this point was also highlighted by Clemens Schannwell. The opening of this paragraph now reads: "The maximum increase in GLF due to the thinning experiments does not equal that of the calving experiments (502% vs 607%), as in the thinning experiments a 1 m thick layer of ice remains across the ice shelf. This means that at the new calving front the ice thickness is linearly interpolated between the unperturbed nodes in the computational mesh and the neighbouring nodes with an ice thickness of 1 m. In the calving experiments mesh elements downstream of the new calving front are removed from the computational domain. We attribute the discrepancy between the maximum GLF increases to this difference in the numerical implementation of the calving and thinning experiments, and not to any residual buttressing effect of the 1 m ice layer."

Line 315: "the maximum basal melt rate was applied across the whole ice shelf" add *with no surface accumulation*

This paragraph has now been removed from the manuscript.

**Anonymous reviewer #2**

**General comment**

I want to thank the authors for having addressed most of my comments during their revisions. The current paper is nicely written and is well detailed. I also appreciate the new simulations they

conducted in the Appendix to better emphasis some arguments from the main text. Yet, my main concern (as for my first review) is the significance of the instantaneous response of ice flow to a mass loss and the lack of a proper treatment of the transient effects following such event. I still think that this drawback limits the interpretation of the experiments, especially in the case of large ice-shelf thinning experiments that should not be treated as an instantaneous event (i.e. large thinning would most likely occur over a long-time scale, giving time to the upstream flow and geometry to adapt).

As the authors argue in their response to reviewers, the aim of this work is to identify the principal buttressing points of Larsen C Ice Shelf and their impact on grounding line flux when this buttressing is released — and, in this regard, the paper is very nicely wrapped. I also understand that the authors are currently conducting transient simulations and that such work can take time. However, I think that the current paper brings only limited insight in comparison to published literature, e.g. Furst et al. (2016) Reese et al. (2018), Gudmundsson et al. (2019) and Zhang et al. (2020) for buttressing, and e.g. Borstad et al. (2017) for the calving of A68.

The main conclusion of the paper, except the quantification of the real calving event of A68, which nicely couples modeling and observations, is that only a small portion of the ice shelf (close to the GL) really matters to buttressing. While this conclusion is in line with pre-cited studies, it does not really bring new insights about buttressing and the methodology is very similar to Reese et al. (2018) and Gudmundsson et al. (2019).

In the current study, the effect of the ice rise seems much smaller than in previous studies. For example, Zhang et al. (2020) also addressed the same question with an adjoint sensitivity (more reproducible and computationally much cheaper, in my opinion) in the context of ocean-induced melting on LCIS. Their results (particularly Fig. 12a) generally agree with Furst et al. (2016). In this regard, the current study results are more in line with Borstad et al. (2013) that see only a small effect of the ice-rise loss on the GLF. However, they observe a 25% change in velocity on the shelf, that, in a transient model, could have important effect. If the authors do not go with transient simulations, I think that they should better relate their paper to the previous literature (comparing results, etc).

For all these reasons, I had a really hard time deciding what recommendation to give for the paper. To me, the lack of transient experiment and the lack of novelty with respect to the published literature are a bit redhibitory. I therefore really recommend to the authors to pursue their effort towards transient simulations and include them in this paper.

We thank the reviewer for their time in providing another thorough review of the manuscript, and their suggestions for further changes that could be made. We hope we have addressed some of these points in this final round of revisions.

We do, however, disagree that our method does not bring new insights into ice-shelf buttressing on the LCIS. Whilst using a similar approach to that of Reese et al. (2018) and Gudmundsson et al. (2019), the diagnostic experiments we conduct allow us to calculate the total buttressing capacity of the ice shelf for the first time, which is not possible using the methods employed in those works and is a conceptual shift in assessing ice-shelf buttressing. It provides a new perspective on how much of the total buttressing different regions of the ice shelf, and the ice rises, provide, rather than assessing GLF sensitivity to ice-shelf perturbations alone.

In regard to our experiments on the loss of basal contact at the ice rises, we agree that our findings are similar to those of Borstad et al. (2013) but note that they do not model the grounded ice, and

therefore cannot calculate the impact on GLF of a loss of contact at the ice rises as we do in this manuscript. We feel that these aspects of our results have been compared to the existing literature in section 4.2, including a comparison with the results presented in the supplementary material of Fürst et al. (2016) on the loss of contact at the Bawden Ice Rise.

Reese et al. (2018) and Zhang et al. (2020) find elevated GLF sensitivity to perturbations in ice-shelf thickness close to the Gipps Ice Rise, but they do not examine the loss of basal contact as we do, and are therefore answer a different question. Similarly, the map of buttressing number of Fürst et al. (2016) shows elevated values upstream of both ice rises, but we take a different conceptual approach in this work. By calculating the total buttressing capacity of the ice shelf, we find that whilst the GLF may be sensitive to perturbations in ice thickness in these regions, the removal of ice here does not significantly impact the GLF when considered against the total buttressing generated by the ice shelf as a whole. We agree that this comparison could be made more explicit and have now included an additional paragraph at the end of section 4.1 which discusses these differences.

**Minor comments**

Title: I would change "ice-shelf thinning" for "thinning" only to avoid a repetition.

This has been addressed in the revised title, which now reads: "The instantaneous impact of calving and thinning on the Larsen C Ice Shelf".

Line 48- 63: I think that I think that the ABUMIP paper from Sun et al. (2020) should be cited in the context of this study.

The following sentence has been added to the introduction on lines 63-64: "Finally, Schannwell et al. (2018) and Sun et al. (2020) explored the transient response of the grounded ice to the complete collapse of the LCIS, and the associated removal of all ice-shelf buttressing."

Line 55: I would change the sentence "Gudmundsson et al. (2019) modelled the impact of an instantaneous perturbation to Antarctic ice-shelf thickness, the spatial pattern and amplitude of which was taken from observations" for "Gudmundsson et al. (2019) modelled the impact of an instantaneous thinning of Antarctic ice-shelf on the grounded ice and GLF, with a pattern and amplitude derived from observations".

Done

Line 58: change "for the last 18 years" for "from 1994 to 2017". I would also reorder the sentence "[...] the instantaneous ice velocity response due to the observed ice-shelf thinning of the last 18 years, and the subsequent reduction in buttressing" as follows "[...] the instantaneous ice velocity response and the reduction in buttressing due to the cumulated observed ice-shelf thinning from 1994 to 2017".

Done

Line 64 to 66: I thank the authors for clearly stating the "instantaneousity" of their experiments. I think that these two sentences could be reshaped in only one, more powerful sentence. For example, "a series of diagnostic perturbation experiments of increasing magnitude" is a bit vague and only gets clearer reading the second sentence. The literature they used to build their experiment is also not only related to LCIS (but the sentence seems to say the opposite, i.e. "existing literature on the ice dynamics of the LCIS").

Thank you for highlighting this, whilst we have not combined these two sentences, the opening sentence of this paragraph has been changed to read: "Here, we build on this existing literature through a series of diagnostic perturbation experiments, including ice-shelf calving and thinning and ungrounding from ice rises."

Line 70: correct the typo: "alnd"

Done

Line 101: Thank you for the addition of Appendix E. Should this be Appendix A, as it is the first you reference in the main text? I do not know what are the referencing rules for Appendix.

We feel that it makes more sense to include this sensitivity study later in the appendices (at least after the appendix on the model initialisation and mesh resolution) but will change this if the journal rules require it.

Line 128: reformat the references in only one parenthesis?

Done

Equation 7 and Appendix A: Why do you choose $\gamma"$#/% = 10)? I agree that your modelled velocities nicely fit observations so that might be a bit picky but the L-curve in Fig A1. (a,b) seems to show that $\gamma"$#/% = 10* is a better value, i.e. with the smallest velocity misfit. Similarly, I would be tempted to say that $\gamma+$#/% = 10 is a better value than 1. It also seems that you treat your L-curves independently but I assume that the choice of one parameter impacts the choice of the others, why not going with a multi-dimension L-curve like in Furst et al. (2016).

The L-curve approach was used to ensure that the regularisation parameters chosen represent an acceptable trade-off between matching observed ice velocities whilst avoiding over-fitting, which is the case for the parameters we used. The values you suggest could also have been chosen, but any parameter combination that lies close to this break in slope of the L-curves is a reasonable choice. A 'multi-dimensional' approach could have been used, but we feel that our parameter choices have been justified by the sensitivity testing presented.

Line 142: penalize deviations?

Done

Line 193: Maybe precise why did you see such negative velocity change. Reorganization of the ice flow due to the change of geometry and buttressing?

This is addressed in the discussion section on lines 254-261.

Line 237: add a comma to "[...] (of 1,500 m), the ice shelf [...]".

Done

Line 238-239: Precise that these metrics are before perturbation.

Done

Line 320: space between "$m$" and "$a./$".

Done

Line 320-324: I think that this is where the instantaneous approach really shows its limit. I don't think that these numbers are really meaningful. When making estimations of future state of the ice sheet, we really want to know what will be the total mass loss, which is not possible with the approach of this paper.

We thank the reviewer for highlighting this and accept their suggestion. We have decided to remove this paragraph from the manuscript.

**Clemens Schannwell**

**General comments:**

The manuscript by Mitcham et al. presents a very thorough and detailed numerical study of the instantaneous effect to a number of idealised ice-shelf thinning and calving perturbations to Larsen C Ice Shelf. For the calving perturbation, they find that most of the buttressing is exerted by floating ice within 5 km downstream of the present-day grounding line position. For the ice-shelf thinning experiments, the authors show that a significant thinning (ca. 200 m) is necessary to get a doubling of ice flux across the grounding line. Overall, I find the manuscript very well written and easy to follow. The provided Figures are appropriate and of high quality. The main criticism in the first round of reviews was about the novelty of the study, as a number of previous studies exist that have investigated the instantaneous response of Larsen C Ice Shelf already, albeit with slightly different foci. I should mention that I was not a reviewer in the first round of reviews.

My opinion is that the depth of experiments including additional sensitivity simulations are just enough to warrant publication as a full research article in TC, without having to undertake transient perturbation simulations. However, I think the fact that this is the instantaneous response should be highlighted throughout. To be fair the authors already do acknowledge this in several places throughout the manuscript. I think that drawing any conclusions about the future should be avoided in the manuscript (e.g. L313-317). In the following I outline my list of minor suggestions below and hope the authors find my comments helpful.

We thank Clemens Schannwell for his time in carrying out this thorough review of the manuscript and his suggestions for further improvements before publication.

**Specific comments:**

• I think the title should already convey the information that this paper is looking at the instantaneous response. Since the paper is also focusing on ice flux across the grounding line, my suggestion would be to also include this in the title. So maybe something along the lines of : "The instantaneous response of Larsen C Ice Shelf grounding line flux to calving and ice-shelf thinning perturbations."

Thank you, we agree that the instantaneous nature of these experiments should be highlighted in the title, but think that including the reference to the grounding line flux makes the title too long, and also misses that fact that we examine the change in ice-shelf velocities in response to the A68 calving and ice-rise ungrounding experiments as well.

Combining this with the suggestion of reviewer #2, the title we have chosen is: "The instantaneous impact of calving and thinning on the Larsen C Ice Shelf".

• Just a comment to line 33-35. It reads like there are no studies on the transient upstream response to changes in Larsen C ice shelf thickness and extent. I would just like to point out that at least for the extreme scenario of complete ice shelf removal, we published a paper in TC in 2018 (Schannwell et al. 2018) about this. We basically find that the sea-level contribution and upstream thinning decay very rapidly (<50 years) after the perturbation.

We have read your paper with interest for the transient experiments that we are conducting – to which it is most relevant – but agree that it should be referenced here. We have included an additional sentence in the introduction that reads: "Finally, Schannwell et al. (2018) and Sun et al. (2020) explored the transient response of the grounded ice to the complete collapse of the LCIS, and the associated removal of all ice-shelf buttressing."

• In section 2.1., can you please mention if you assume isothermal ice and if so what temperature and why?

There is no thermodynamic component in Úa, and therefore the temperature of the ice does not enter the system directly. It will of course impact the rate factor, and we discuss how we choose a prior value for that – based on a uniform ice temperature – in section 2.3. The impact of any variation in ice temperature on the flow is implicitly captured in the rate factor during the model initialisation.

• In section 2.2 L107-108, this reads like Gmsh is creating linear finite elements. As far as I know, Gmsh simply creates the triangulation or mesh, but you can use any finite element type on this mesh. Does this mean that you are using linear Lagrange elements? Would be nice to spell this out more explicitly.

Thank you for highlighting this. We have made this more explicit, and the opening of this paragraph now reads: "The finite element mesh used in the computation was generated with the open source Gmsh software (Geuzaine and Remacle, 2009). We chose to use linear shape functions on these elements."

• L305-307, I find the result that a 1 m or even 0.001 m thick ice shelf vs. no ice shelf gives a difference of ~100% in ice flux across the grounding line surprising. This warrants a discussion why there is such a large discrepancy.

We have examined this again and find that this discrepancy is due to the different numerical implementation of the two experiment types. We have now expanded this discussion and the opening of this paragraph now reads: "The maximum increase in GLF due to the thinning experiments does not equal that of the calving experiments (502% vs 607%), as in the thinning experiments a 1 m thick layer of ice remains across the ice shelf. This means that at the new calving front the ice thickness is linearly interpolated between the unperturbed nodes in the computational mesh and the neighbouring nodes with an ice thickness of 1 m. In the calving experiments mesh elements downstream of the new calving front are removed from the computational domain. We attribute the discrepancy between the maximum GLF increases to this difference in the numerical implementation of the calving and thinning experiments, and not to any residual buttressing effect of the 1 m ice layer."

• L313-316 I find this scenario to be a bit too far fetched and recommend deleting this paragraph.

Thank you, this was also highlight by the second anonymous reviewer. We agree and have now removed this paragraph from the manuscript.

• Appendix E: I do not know if Ua has pressure-limited sliding laws implemented (Tsai, Budd, or Schoof sliding relation), but I think as a community we are moving towards pressure-limited sliding relations, so it would be interesting to see if the effect between these two relations is already large in the instantaneous response or only in transient simulations.

We feel that further sensitivity testing of these results with multiple sliding laws is beyond the scope of this appendix. But pressure-dependent sliding laws have now been implemented in Úa and could be tested in future work.

• Appendix E: Why did you rerun the inversion for different sliding law exponents? In theory you should be able to use one inversion for all different exponents because they must satisfy C2|u|m2 = C1|u|m1, where m1 = 1 and m2 = 2 for example. You could then rearrange that for C2. Maybe you can comment on that?

We are aware of this approach and in fact often do exactly this to arrive at starting values for our A and C distributions. But this is not exactly a correct solution to the inverse problem (correct in the sense that it provides a minimum point) because the cost functions we use also involve A and C directly in the regularisation terms. The method suggested by the reviewer would work just fine if the only term in our cost function was the misfit term (i.e. the likelihood), but as we have these additional regularisation terms we always find that further inverse iterations are required.

Technical corrections:

L69 outlined and labelled

Done

L86 I think this should be reworded. The stress balance equation is always solved diagnostically. There is no time-dependence in Eq. 1. Only when you couple the stress equation to a transport equation (e.g. ice-thickness evolution equation) does it become transient (time dependent).

Thank you for highlighting this, it has been re-worded for clarity on this point and now reads: "In this work we conduct diagnostic – or time-independent – experiments, in which the equations for stress balance are solved together …"

L430 Whilst

Done

Figures:

Fig. 1: Does the ps in the axes label stand for polar stereographic? I would consider scratching this.

We would prefer to keep the ps reference in all map axes in the manuscript, but will make this explicit in the caption to Fig. 1.

Fig. 5: Is there really only speed-up in panels a-c?

There is almost exclusively speed-up in those three experiments, and the small decrease in speed that was modelled over a few disparate nodes within the computational mesh (almost entirely < 10 m/a) was not visible when plotted. We therefore decided to truncate the colour scale at 0 for clarity. We thank the reviewer for raising this point and will add a sentence in the caption to explain this.

Fig. C1: sizes were half

Done

Fig. E1: (Eq.5).

Done

Sincerely,

Clemens Schannwell

References

Schannwell, C., Cornford, S., Pollard, D., and Barrand, N. E.: Dynamic response of Antarctic Peninsula Ice Sheet to potential collapse of Larsen C and George VI ice shelves, The Cryosphere, 12, 2307–2326, https://doi.org/10.5194/tc-12-2307-2018, 2018